# Allosteric activation of the co-receptor BAK1 by the EFR receptor kinase initiates immune signaling

**Henning Mühlenbeck[1], Yuko Tsutsui[2,3], Mark A Lemmon[2,3], Kyle W Bender[1]\*, Cyril Zipfel[1,4]\***

[1]Institute of Plant and Microbial Biology, Zürich-Basel Plant Science Center, University of Zürich, Zürich, Switzerland; [2]Department of Pharmacology, Yale University School of Medicine, New Haven, United States; [3]Yale Cancer Biology Institute, Yale University West Campus, West Haven, United States; [4]The Sainsbury Laboratory, University of East Anglia, Norwich Research Park, Norwich, United Kingdom

**\*For correspondence:**
kyle.bender@botinst.uzh.ch (KWB);
cyril.zipfel@botinst.uzh.ch (CZ)

**Competing interest:** The authors declare that no competing interests exist.

**Abstract** Transmembrane signaling by plant receptor kinases (RKs) has long been thought to involve reciprocal trans-phosphorylation of their intracellular kinase domains. The fact that many of these are pseudokinase domains, however, suggests that additional mechanisms must govern RK signaling activation. Non-catalytic signaling mechanisms of protein kinase domains have been described in metazoans, but information is scarce for plants. Recently, a non-catalytic function was reported for the leucine-rich repeat (LRR)-RK subfamily XIIa member EFR (elongation factor Tu receptor) and phosphorylation-dependent conformational changes were proposed to regulate signaling of RKs with non-RD kinase domains. Here, using EFR as a model, we describe a non-catalytic activation mechanism for LRR-RKs with non-RD kinase domains. EFR is an active kinase, but a kinase-dead variant retains the ability to enhance catalytic activity of its co-receptor kinase BAK1/SERK3 (brassinosteroid insensitive 1-associated kinase 1/somatic embryogenesis receptor kinase 3). Applying hydrogen-deuterium exchange mass spectrometry (HDX-MS) analysis and designing homology-based intragenic suppressor mutations, we provide evidence that the EFR kinase domain must adopt its active conformation in order to activate BAK1 allosterically, likely by supporting αC-helix positioning in BAK1. Our results suggest a conformational toggle model for signaling, in which BAK1 first phosphorylates EFR in the activation loop to stabilize its active conformation, allowing EFR in turn to allosterically activate BAK1.

## eLife assessment

This manuscript reports **important** in vitro biochemical and in planta experiments to study the receptor activation mechanism of plant membrane receptor kinase complexes through the non-catalytic function of an active protein kinase. Several lines of evidence **convincingly** show that one such receptor kinase with pseudokinase-like function, the immune receptor EFR achieves an active conformation following phosphorylation by a co-receptor kinase, and then in turn activates the co-receptor kinase allosterically to enable it to phosphorylate down-stream signaling components. This manuscript will be of interest to scientists focusing on cell signalling and allosteric regulation.

## Introduction

Plants and metazoans respond to extracellular signals through different sets of plasma membrane receptors. Whereas the G-protein coupled receptor family expanded in metazoans, the receptor

kinase (RK) and receptor protein (RP) families expanded in plants (*Shiu and Bleecker, 2003*; *Shiu and Bleecker, 2001*; *Trusov and Botella, 2016*). Plant RKs resemble metazoan receptor tyrosine kinases (RTKs) in their organization, containing an extracellular ligand-sensing domain that is coupled to an intracellular kinase domain by a single-pass transmembrane helix (*Hohmann et al., 2017*; *Lemmon and Schlessinger, 2010*). In contrast to RTKs, however, the intracellular kinase domain of plant RKs is a serine/threonine kinase domain most closely related to interleukin-1 receptor associated kinases (IRAKs)/Pelle kinases (*Shiu and Bleecker, 2003*; *Shiu and Bleecker, 2001*) – although dual-specificity kinase activity has been reported in some cases (*Liu et al., 2018*; *Luo et al., 2020*; *Macho et al., 2014*; *Oh et al., 2009*; *Perraki et al., 2018*).

The leucine-rich repeat (LRR)-RKs are the most extensively studied sub-family of *Arabidopsis thaliana* (hereafter *Arabidopsis*) RKs. They primarily sense peptide ligands that regulate growth and development, or molecular patterns that are released during damage or microbial infection (*Couto and Zipfel, 2016*). Ligand-binding LRR-RKs have long ectodomains (>20 LRRs), and recruit short LRR ectodomain-containing co-receptor kinases (coRKs) upon ligand perception (*Bender and Zipfel, 2023*; *Hohmann et al., 2017*), most of which belong to the SERK (somatic embryogenesis receptor kinase) family. One of the most well characterized LRR-RKs is BRI1 (brassinosteroid insensitive 1), which recruits the coRKs SERK1 and SERK3/BAK1 (BRI1-associated kinase 1) upon brassinosteroid (BR) perception (*Albrecht et al., 2008*; *He et al., 2000*; *Hothorn et al., 2011*; *Nam and Li, 2002*; *Santiago et al., 2013*). After ligand perception, the intracellular kinase domains of BRI1 and BAK1 reciprocally trans-phosphorylate each other in their activation loops (*Hohmann et al., 2017*), and BAK1 then phosphorylates BRI1 in its juxtamembrane segment and C-tail to propagate BR signaling (*Wang et al., 2008*; *Wang et al., 2005*). BRI1 exemplifies a set of BAK1-dependent LRR-RKs that have RD-type intracellular kinase domains and require their catalytic activity to signal (*Cao et al., 2013*; *Kosentka et al., 2017*; *Taylor et al., 2016*). Other LRR-RKs such as EFR (elongation factor Tu receptor) instead have non-RD intracellular kinases, and are thought not to *trans*-phosphorylate their associated coRK following ligand perception (*Schwessinger et al., 2011*). Moreover, kinase inactive mutants of EFR (D849N or K851E) retain signaling function, a result that challenges the generality of the reciprocal trans-phosphorylation model (*Bender et al., 2021*).

EFR perceives the pathogen-associated molecular pattern (PAMP) elongation factor Tu, or its active peptide epitope elf18 (*Zipfel et al., 2006*). PAMP perception triggers heterodimerization with the coRK BAK1, resulting in phosphorylation of the EFR intracellular kinase domain (*Bender et al., 2021*; *Roux et al., 2011*; *Schulze et al., 2010*; *Schwessinger et al., 2011*). The signal is subsequently relayed to the cytoplasmic kinases BIK1 (botrytis-induced kinase 1) and PBL1 (PBS1-like 1) (*Li et al., 2014*; *Lu et al., 2010*; *Ranf et al., 2014*). The resulting immune signaling activation elicits a battery of cellular responses, including an apoplastic oxidative burst (hereafter oxidative burst), $Ca^{2+}$-influx, callose deposition, MAPK (mitogen activated protein kinase) activation, and transcriptional reprogramming (*DeFalco and Zipfel, 2021*).

Initial studies in vitro identified phosphorylation sites in EFR that result either from autophosphorylation or from trans-phosphorylation by BAK1 (*Wang et al., 2014*). More recently, in vivo phosphorylation sites on EFR were identified by immunoprecipitating EFR-GFP from elf18-treated seedlings (*Bender et al., 2021*). One was a serine (S888) in the activation loop (A-loop), at which phosphorylation was consistently observed in vitro and in vivo (*Bender et al., 2021*; *Wang et al., 2014*). Despite the non-RD nature of EFR, and its ability to signal independently of catalytic activity, ligand-inducible phosphorylation in the A-loop (S887/S888) surprisingly proved indispensable for signaling (*Bender et al., 2021*). Furthermore, a functionally important tyrosine in EFR (Y836) – conserved in subdomain VIa of many eukaryotic protein kinases (*Lai et al., 2016*; *Luo et al., 2020*; *Perraki et al., 2018*) – was found to be phosphorylated after ligand treatment in vivo (*Macho et al., 2014*). A phospho-ablative Y836F mutation in EFR blocks immune signaling and resistance against the phytopathogenic bacterium *Pseudomonas syringae* (*Macho et al., 2014*). Thus, although EFR function does not require its catalytic activity, both A-loop phosphorylation and phosphorylation of the VIa subdomain tyrosine (VIa-Tyr) appear to be crucial. The mechanistic importance of these phosphorylation events remains poorly understood, but we hypothesized that they switch the EFR kinase domain into an active-like conformation that allosterically activates BAK1 in the EFR-BAK1 complex through a mechanism similar to that described for pseudokinases (*Mace and Murphy, 2021*; *Sheetz and Lemmon, 2022*) – ultimately promoting BAK1's activity toward its substrate BIK1.

Here, we tested the hypothesis that EFR is an allosteric regulator of BAK1 using a range of different approaches. Our in vitro studies revealed that forcing dimerization of the EFR and BAK1 intracellular domains allosterically enhances BAK1 activity. Using a homology-guided approach, we designed mutations to stabilize the active-like conformation of the EFR kinase domain and found that they restore functionality of EFR variants that cannot be phosphorylated in the A-loop (EFR$^{SSAA}$) or at Y836 (EFR$^{Y836F}$). We also used hydrogen-deuterium exchange mass spectrometry (HDX-MS) to analyze conformational dynamics, revealing that the Y836F mutation hampers the ability of the EFR kinase domain to adopt an active-like conformation. Collectively, our findings argue that the active conformation of the EFR kinase domain is required for allosteric activation of the BAK1 kinase domain. Finally, we present evidence suggesting that EFR activates BAK1 allosterically by supporting αC-helix positioning in BAK1.

## Results

### The EFR intracellular domain allosterically activates BAK1 in vitro

To test the hypothesis that EFR enhances BAK1 catalytic activity allosterically, we used rapamycin (Rap)-induced dimerization (RiD) to induce a complex between the isolated EFR and BAK1 intracellular domains (*Banaszynski et al., 2005*; *Kim et al., 2021*). The RiD system was previously applied in planta, maintaining membrane-association by N-terminal myristoylation (*Kim et al., 2021*). For our in vitro experiments, the myristoylation sites were excluded to facilitate the purification of recombinant protein. We tested two LRR-RKs: EFR and BRI1, fusing GFP C-terminally and FKBP (FK506 binding protein) N-terminally to their intracellular domains. BAK1 instead was N-terminally tagged with FRB (FKBP-Rap-binding). We first confirmed that adding Rap induced formation of FKBP-EFR/FRB-BAK1 dimers in size exclusion chromatography experiments (*Figure 1A*). We then assessed the effect of inducing RK/coRK kinase complex formation on their ability to phosphorylate BIK1$^{D202N}$. BIK1 was chosen as it is a reported substrate of both, EFR/BAK1 and BRI1/BAK1 complexes (*Lin et al., 2013*). Rap addition increased BIK1$^{D202N}$ phosphorylation when the BRI1 or EFR kinase domains were dimerized with BAK1 (*Figure 1B and C*). Kinase-dead variants with the catalytic residue (HRD-aspartate) replaced by asparagine (EFR$^{D849N}$ and BRI1$^{D1009N}$), had distinct effects. BRI1$^{D1009N}$ failed to enhance BIK1 phosphorylation substantially, whereas EFR$^{D849N}$ retained some ability to do so (*Figure 1B and C*). The same trend was observed for phosphorylation of the BAK1 kinase domain itself, indicating that EFR also enhances BAK1 autophosphorylation activity (*Figure 1B and C*).

The increased BIK1 trans-phosphorylation observed in these in vitro RiD experiments could arise either from direct enhancement of BAK1 activity by the RK kinase domain or from more efficient BIK1$^{D202N}$ recruitment to the dimerized RK/coRK complex. We found that BIK1$^{D202N}$ did not co-elute with the EFR-BAK1 complex in size exclusion studies, indicating that stable EFR/BAK1/BIK1 trimers do not form in vitro (*Figure 1A*). Our findings therefore support the hypothesis that EFR increases BIK1 phosphorylation by allosterically activating the BAK1 kinase domain. Although EFR's catalytic activity is dispensable for this effect – and for immune signaling – EFR must be phosphorylated at S887/S888 in the A-loop and Y836 to signal in vivo (*Bender et al., 2021*; *Macho et al., 2014*). The phospho-ablative EFR A-loop mutant EFR$^{SSAA}$ fails to activate the RK/coRK complex, as ligand-induced phosphorylation of BAK1 S612 – a mark for active BAK1-containing receptor complexes (*Perraki et al., 2018*) – is obstructed despite retaining the ability to associate with BAK1 in a ligand-dependent manner (*Bender et al., 2021*). To test whether the requirement for Y836 phosphorylation is similar, we immunoprecipitated EFR-GFP and EFR$^{Y836F}$-GFP from mock- or elf18-treated seedlings and probed co-immunoprecipitated BAK1 for S612 phosphorylation. EFR$^{Y836F}$ also obstructed the induction of BAK1 S612 phosphorylation (*Figure 1—figure supplement 1*), indicating that EFR$^{Y836F}$ and EFR$^{SSAA}$ impair receptor complex activation.

### EFR VIa-Tyr mutation affects dynamics of regulatory kinase subdomains

Y836 in EFR corresponds in sequence and structural alignments with Y156 in the canonical kinase PKA (protein kinase A) (*Figure 2—figure supplements 1 and 2A*), which is conserved in many protein kinases. We therefore speculated that Y836 may function as a pivot for key movements of the regulatory αC-helix, controlling conformational toggling of inactive/active transitions as described for its PKA counterpart (*Tsigelny et al., 1999*). To test whether an EFR Y836F mutation in EFR interferes

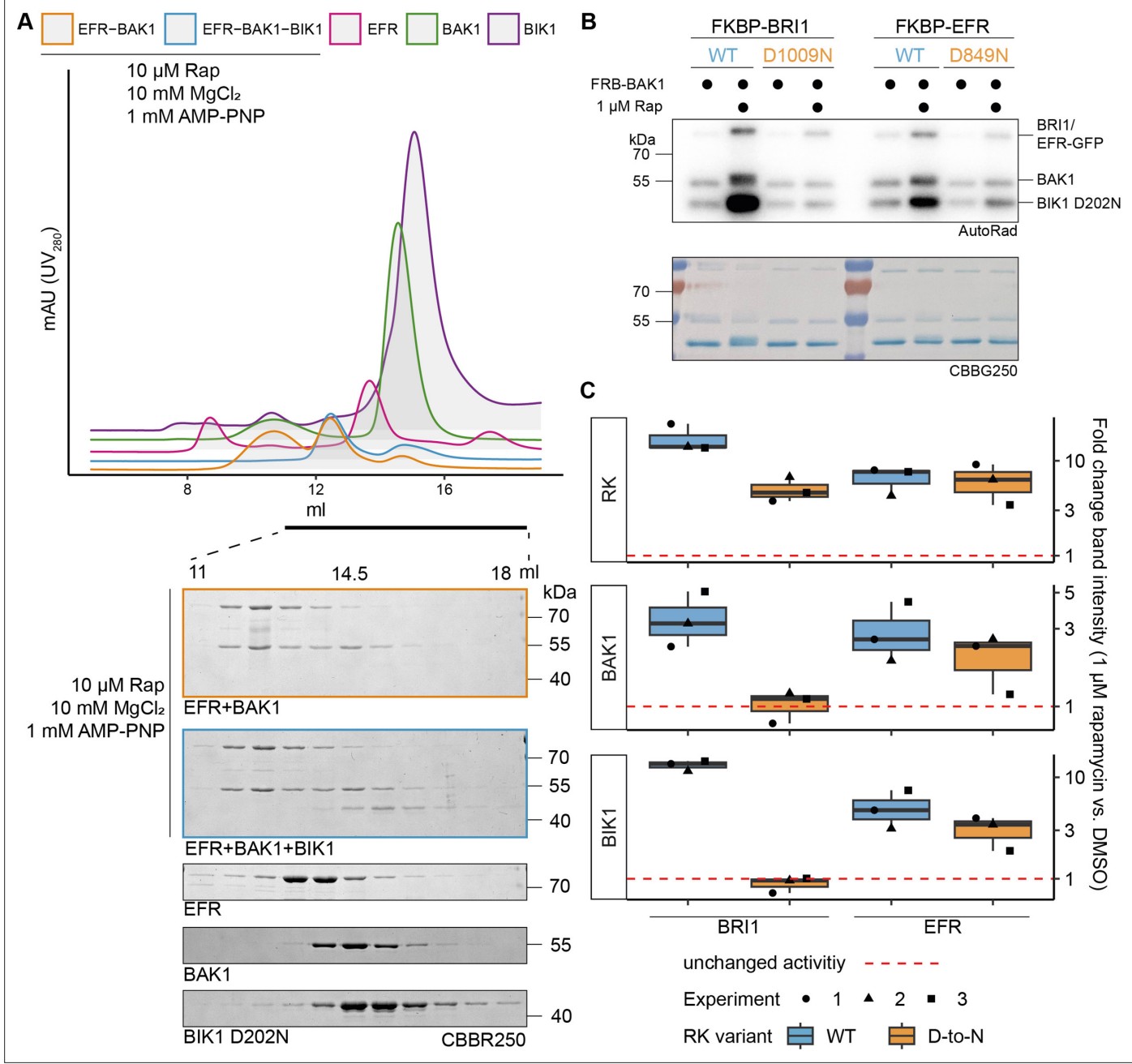

**Figure 1.** EFR facilitates BIK1 trans-phosphorylation by BAK1 non-catalytically. The kinase domains of coRK BAK1 and the RKs BRI1 and EFR were tagged with RiD domains and purified from *E. coli* λ PP cells. (**A**) Recombinantly expressed RiD-tagged kinase domains were mixed together at equimolar ratios (2 µM), with or without addition of 10 µM Rap as well as 10 mM MgCl₂ and 1 mM AMP-PNP. (**B**) RK and coRK were mixed at an equimolar ratio at 50 nM, kinase-dead BIK1$^{D202N}$ substrate was added at 500 nM. Reactions were carried out at RT for 10 min with 0.5 µCi [γ-³²P] ATP, 100 µM ATP and 2.5 mM each of MgCl₂ and MnCl₂. Addition of 1 µM Rap enhanced transphosphorylation of BIK1 by EFR and BRI1. Kinase-dead BRI1 failed to enhance BIK1 transphosphorylation, but kinase-dead EFR retained some ability to do so. A similar trend was observed for (auto) phosphorylation of BAK1 itself. (**C**) Quantification of band intensities over three independent experiments of which a representative is shown in B.

The online version of this article includes the following source data and figure supplement(s) for figure 1:

**Source data 1.** Raw data for gel images in *Figure 1A*.

**Source data 2.** Raw data for autoradiography in *Figure 1B*.

**Figure supplement 1.** EFR$^{Y836F}$ compromises ligand-induced receptor complex activation.

**Figure supplement 1—source data 1.** Raw data for immunoblots in *Figure 1—figure supplement 1*.

with assembly of the active-like EFR kinase conformation we used HDX-MS. Plotting differential HDX (Δ%EX) between unphosphorylated EFR and EFR[Y836F] (aa684-1031) identifies one region that is stabilized in EFR[Y836F] compared to wild-type and two that are destabilized (*Figure 2A and B*). The region stabilized in EFR[Y836F] contains the N-terminal part of the A-loop, implying that this region makes more extensive contacts with the kinase core than in the unphosphorylated wild-type kinase domain. Alternatively, the decreased deuterium uptake in the EFR[Y836F] A-loop could arise from stabilization of a short A-loop α-helix, as seen in the inactive conformation of many kinases (such as in epidermal growth factor receptor). The two regions that become more structurally flexible in EFR[Y836F] include the β3-αC loop and the catalytic loop plus C-terminal end of the αE-helix – where Y836 is located (*Figure 2A and B*). Because these two regions are both important for conformational switching, the HDX-MS results suggest that Y836 is important in regulating kinase domain allosteric transitions. Consistent with this, crystal structures as well as AlphaFold2 models of active kinase conformations (*Faezov and Dunbrack, 2023*) suggest that the side-chain of this tyrosine forms hydrogen-bonds (H-bond) with the αC-β4 loop backbone to establish an inter-lobe connection (*Figure 2—figure supplement 1*). Loss of this inter-lobe connection may underlie the observed alterations in EFR[Y836F] conformational dynamics in EFR[Y836F], resulting from inaccessibility of the active-like conformation.

## Kinase activating mutations restore partial function of EFR[Y836F] and EFR[SSAA]

To build on these results, we next sought to rescue the active-like kinase conformation of EFR[Y836F] by introducing activating mutations. Kinase activating mutations are well known in human disease, and cause different malignancies (*Foster et al., 2016*; *Hu et al., 2015*). Because the HDX-MS data indicated a destabilized αC-helix, we were specifically interested in activating mutations thought to stabilize the kinase αC-helix in a 'swung-in' state, with the goal of making homologous changes in EFR. Such mutations were systematically identified in BRAF/CRAF by searching oncogenes using phenylalanine substitutions (*Hu et al., 2015*). We exploited these oncogenic BRAF mutations for homology-based design of putative activating EFR mutations at corresponding positions that could function as intragenic suppressors of EFR[Y836F] (*Table 1* and *Figure 2—figure supplement 2A and B*). Since EFR already carries a phenylalanine at the position corresponding to L505 in BRAF, we generated EFR[F761[H/M]] to resemble mutations known to potently activate BRAF (*Hu et al., 2015*).

The putative activating mutations were introduced into either EFR[WT] or EFR[Y836F] and transiently expressed in *Nicotiana benthamiana* to test receptor function. As expected, heterologous expression of EFR[WT] but not of EFR[Y836F] conferred elf18 sensitivity (*Figure 2B*, *Figure 2—figure supplement 2C*). When introduced on their own into EFR[WT], the single mutations L743F, F761[HM], and L873E were all consistent with wild-type EFR function, supporting an elf18-induced oxidative burst, except for the ΔNLLKH deletion (*Figure 2—figure supplement 2C*). Intriguingly, the F761[H/M] mutations were also able to partially restore the ability of EFR[Y836F] to support an elf18-induced oxidative burst (*Figure 2B* and *Figure 2—figure supplement 2C*). This observation also extended to EFR[SSAA] (*Figure 2C*), arguing that the F761[H/M] mutations support the ability of EFR to trans-activate BAK1.

EFR A-loop phosphorylation was reported previously to be indispensable for EFR function, and led to the hypothesis that it controls conformational switching, despite EFR being a non-RD kinase (*Bender et al., 2021*). Our finding that the F761H mutation restores EFR[SSAA] function further supports this hypothesis. Nevertheless, how EFR A-loop phosphorylation facilitates conformational switching is unclear. In our AlphaFold2 models of the EFR kinase domain, we noted that two basic residues from the β3-αC loop and the αC-helix extend toward the A-loop and, similar to PKA H87 (*Meharena et al., 2016*), may coordinate A-loop phosphorylation (*Figure 2—figure supplement 3*).

## EFR F761H restores function of EFR[Y836F] and EFR[SSAA] in *Arabidopsis*

We next generated stable *Arabidopsis* complementation lines expressing *pEFR::EFR(variant)-GFP::HSP18t* constructs in the null *efr-1* background. Because the F761H mutation showed greater ability than F761M to rescue function of both EFR[Y836F] and EFR[SSAA] (*Figure 2B and C*), we chose this secondary mutation for the generation of complementation lines. Two independent homozygous complementation lines were isolated, with similar accumulation of EFR-GFP protein in the T3 generation. For EFR[F761H/Y836F], only one homozygous T3 line could be isolated, so a heterozygous T2 line with two T-DNA insertion events was used for physiological experiments. Similar to experiments in *N.*

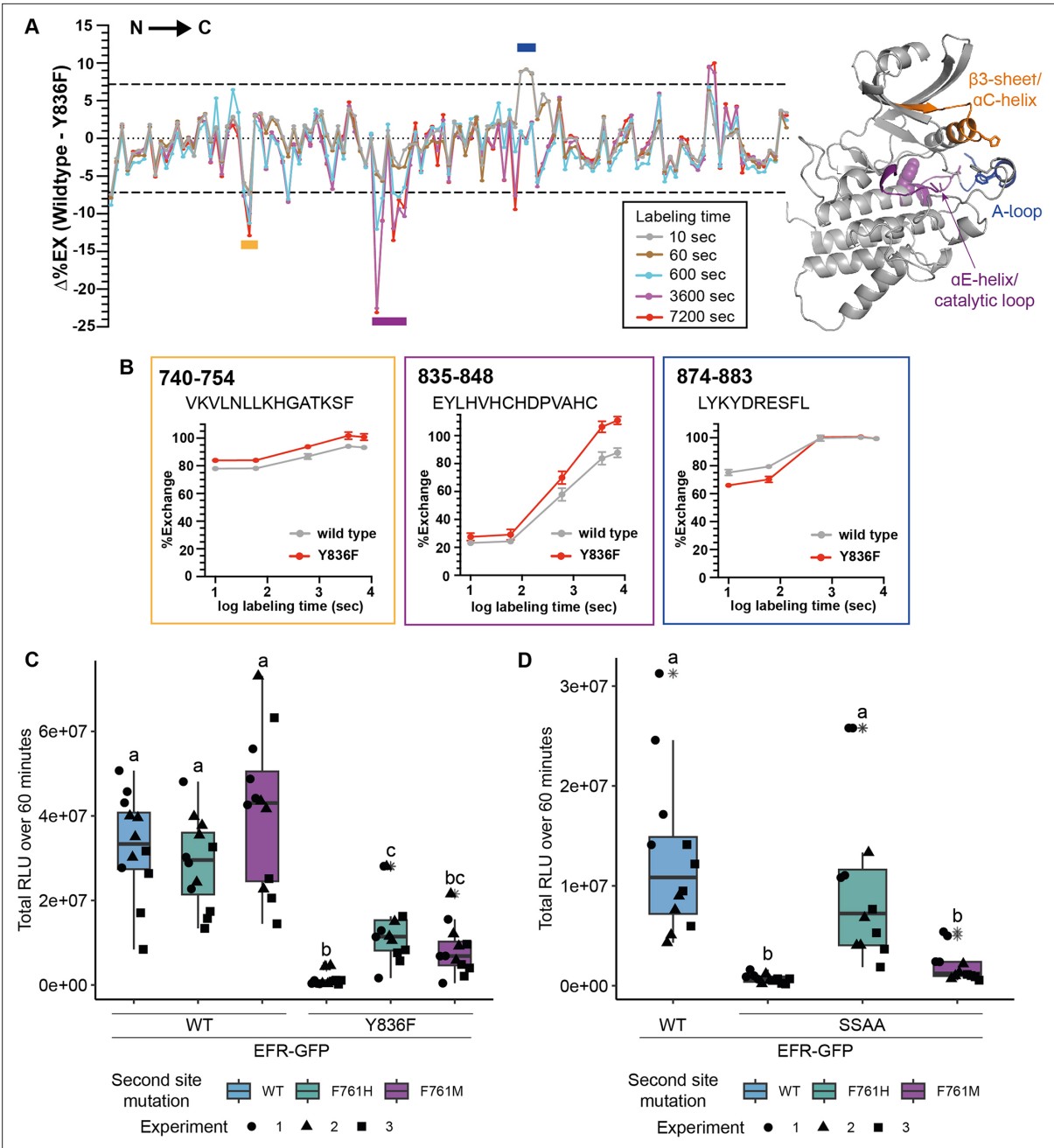

**Figure 2.** EFR[Y836F] and EFR[SSAA] impair the active kinase conformation, which is required for signaling function. (**A**) (left) HDX-MS results for unphosphorylated EFR and EFR[Y836F] protein. The difference in percent H/D exchange in wild type EFR and EFR[Y836F] is expressed as the Δ%EX (wild type EFR – EFR[Y836F]), with the positive and negative Δ%EX indicating more stabilized and destabilized regions in EFR[Y836F], respectively, compared to wild-type EFR. The Δ%EX values at different labeling time points are shown as colored lines, as indicated in the figure. The horizontal dotted black lines indicate the 98% confidence interval for the Δ%EX data (±7.18%, corresponding to ±0.4 Da difference between wild type and Y836F percent exchange) calculated as described previously (*Houde et al., 2011*). Regions with Δ%EX values that exceed this confidence limit are indicated as colored bars in the figure, including the β3-αC loop (orange), the catalytic loop plus part of αE (purple), and the A-loop (blue). These regions are colored in the AlphaFold2-derived model of the EFR kinase domain shown at right, in which Y836 is shown as a purple sphere. All data are the average of three independent biological repeats (n=3) with three technical repeat experiments each. A summary of the HDX-MS analysis is presented in *Table 3*. (**B**) HDX-MS analysis of representative peptides from regions with significantly different HD exchange. Frames are color-coded according to regions in A. Amino acid range of the peptides in full length EFR are indicated in the top left corner and the sequence below. (**C, D**) Secondary site mutation EFR F761[H/M] partially restores function of EFR[Y836F] (**C**) and EFR[SSAA] (**D**). Full length EFR and its variants were expressed transiently in *N. benthamiana* and their function was tested in an oxidative burst assay. EFR F761H partially restored oxidative bursts of EFR[Y836F] and EFR[SSAA]. Outliers are in indicated by asterisk in addition to

*Figure 2 continued on next page*

*Figure 2 continued*

the outlier itself and are included in statistical analysis; Statistical test: Kruskal-Wallis test ($p<2.2*10^{-16}$ in C, $p=1.163*10^{-7}$ in D), Dunn's post-hoc test with Benjamin-Hochberg correction ($p \leq 0.05$) Groups with like lowercase letter designations are not statistically different.

The online version of this article includes the following figure supplement(s) for figure 2:

**Figure supplement 1.** VIa-Tyr forms H-bonds with the αC-β4 loop in various predicted and solved structures.

**Figure supplement 2.** Rational design of activating mutations in EFR and screen for functional recovery of EFR$^{Y836F}$.

**Figure supplement 3.** EFR A-loop phosphorylation sites may coordinate with basic residues from the β3-αC loop and αC-helix.

*benthamiana*, the oxidative burst was partially restored in stable *Arabidopsis* complementation lines that express EFR$^{F761H/Y836F}$ or EFR$^{F761H/SSAA}$ (*Figure 3—figure supplement 1A*).

The oxidative burst is only one of multiple cellular responses triggered by elf18. We therefore also tested whether other immune signaling responses are restored in the EFR$^{F761H/Y836F}$ and EFR$^{F761H/SSAA}$ complementation lines. Consistent with previous results, EFR$^{Y836F}$ and EFR$^{SSAA}$ lines were both less sensitive to treatment with 5 nM elf18 than those with wild-type EFR in seedling growth inhibition (SGI) assays (*Bender et al., 2021*; *Macho et al., 2014*; *Figure 3—figure supplement 1B*). As expected, both independent complementation lines of EFR$^{F761H/Y836F}$ and EFR$^{F761H/SSAA}$ exhibited enhanced SGI compared to EFR$^{Y836F}$ and EFR$^{SSAA}$, respectively (*Figure 3—figure supplement 1B*). We also observed that MAPK activation was abolished or severely impaired in both EFR$^{Y836F}$ and EFR$^{SSAA}$ complementation lines (*Figure 3—figure supplement 1C*) and recovered in EFR$^{F761H/Y836F}$ and EFR$^{F761H/SSAA}$ complementation lines (*Figure 3—figure supplement 1C*).

The recovery of multiple immune responses in EFR$^{F761H/Y836F}$ and EFR$^{F761H/SSAA}$ complementation lines suggested that they retain fully functional elf18 signaling and that effective resistance against bacteria can be established. To confirm this, we tested the transgenic lines for resistance against *Agrobacterium tumefaciens*. Infection by this bacterium is restricted in *Arabidopsis* by EFR, so loss-of-function mutants like *efr-1* are more susceptible (*Zipfel et al., 2006*). *A. tumefaciens* carrying a plasmid with an intronic version of the *β-glucuronidase* (*GUS*) gene was used to evaluate infection success. GUS activity in plant protein extracts following infection correlates with the ability of the bacteria to transiently transform plant cells. Significant GUS activity was detected in *efr-1* (*Figure 3A*), consistent with successful *A. tumefaciens* infection, but only little was seen in wild-type EFR complementation lines, reflecting resistance to infection (*Bender et al., 2021*; *Zipfel et al., 2006*). The EFR$^{Y836F}$ and EFR$^{SSAA}$ complementation lines were both more susceptible to *A. tumefaciens* transformation than wild-type EFR complementation lines, as indicated by elevated GUS activity (*Figure 3A*), but this was greatly diminished in the EFR$^{F761H/Y836F}$ and EFR$^{F761H/SSAA}$ complementation lines. Hence, these experiments show that the EFR F761H mutation restores full signaling function of EFR$^{Y836F}$ and EFR$^{SSAA}$ and thus resistance against *A. tumefaciens*.

**Table 1.** Homology-based design of putative intragenic suppressor mutations for EFR.
The list contains the residue number of EFR and the analogous oncogenic mutation in BRAF, as well as a short description of the mode of action of the oncogenic mutation. See *Figure 2—figure supplement 1B* for structural locations.

| EFR mutation | Analogous oncogenic mutation | Mode of action | Source |
|---|---|---|---|
| L743F | BRAF L485F | Extended hydrophobic interaction network along the αC-helix | *Hu et al., 2015* |
| F761[H/M] | BRAF L505[F/H/M] | Enforcement of hydrophobic interactions in the regulatory spine | *Hu et al., 2015* |
| ΔNLLKH | BRAF ΔNVTAP | Shortening of β3-αC-helix loop, pulling 'in' the αC-helix | *Foster et al., 2016* |
| L873E | BRAF V600E | Upward bending of C-terminal αC-helix end | *Hu et al., 2015* |

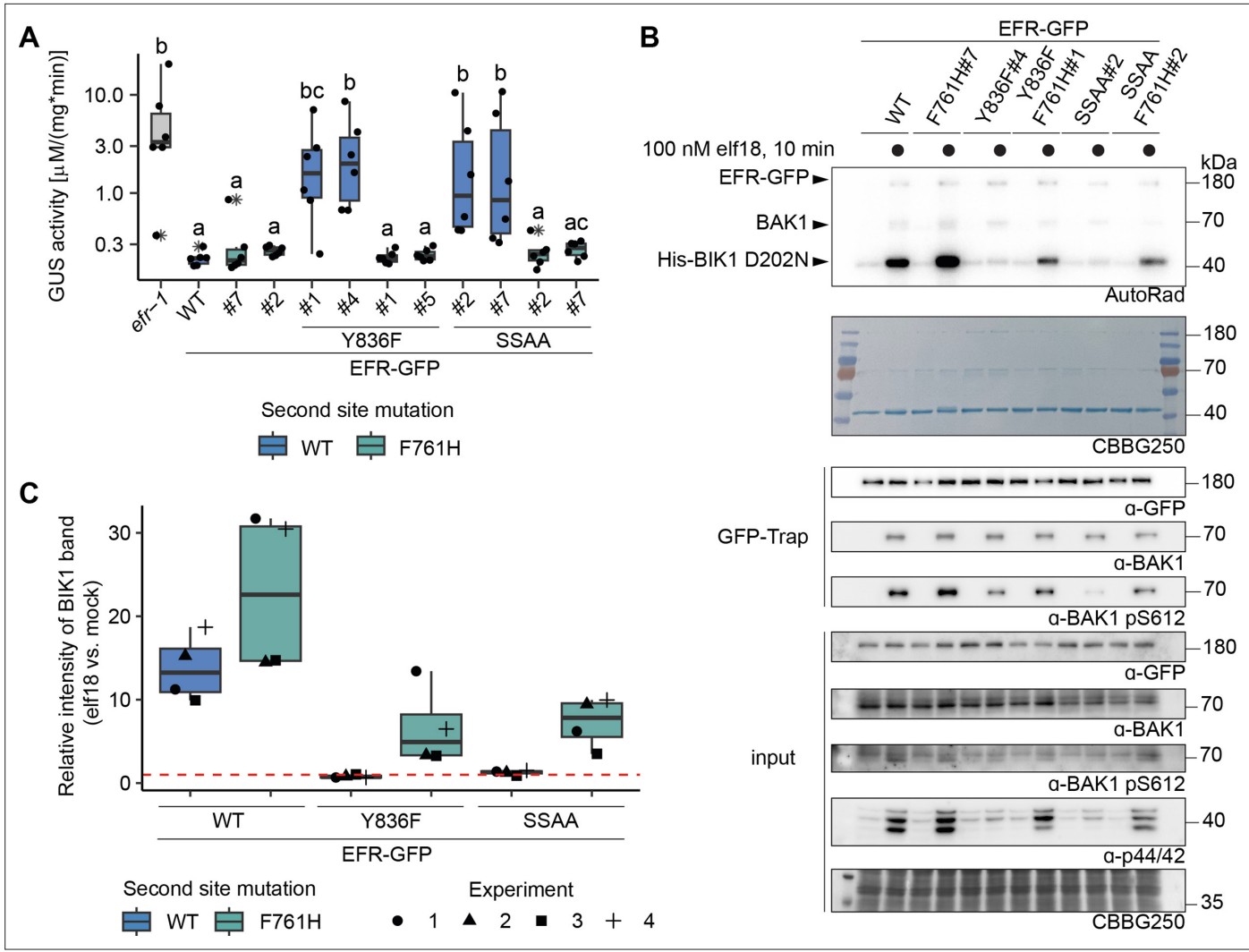

**Figure 3.** EFR^F761H/Y836F and EFR^F761H/SSAA recover receptor complex activation. (**A**) In infection assays, GUS activity was high in the positive control *efr-1* line. GUS activity level was reduced in the EFR^WT and EFR^F761H complementation lines, but much less so in the EFR^Y836F and EFR^SSAA complementation lines. By contrast, EFR^F761H/Y836F and EFR^F761H/SSAA complementation lines displayed substantially repressed GUS activity. Each experiment was repeated three times with similar results. Outliers are indicated by an additional asterisk and included in statistical analysis. Statistical test: Kruskal-Wallis test ($P=5.704*10^{-7}$), Dunn's post-hoc test with Benjamin-Hochberg correction ($P \leq 0.05$) Groups with like lowercase letter designations are not statistically different. (**B**) In IP kinase assays, ligand-induced interaction of EFR^WT and EFR^F761H with BAK1 increased transphosphorylation of BIK1^D202N, but this was abolished for EFR^Y836F and EFR^SSAA. Both EFR^F761H/Y836F and EFR^F761H/SSAA showed partially restored BIK1^D202N trans-phosphorylation as well as BAK1 S612 phosphorylation (across four replicates for EFR^F761H/SSAA and in two out of four replicates for EFR^F761H/Y836F). Samples were also probed for MAPK phosphorylation for effective ligand treatment. Treatment: 100 nM elf18 for 10 min. (**C**) Quantification of BIK1^D202N band intensity observed in autoradiographs from the four independent replicates performed. Dotted red line indicates unchanged band intensity in mock vs. elf18 treatment.

The online version of this article includes the following source data and figure supplement(s) for figure 3:

**Source data 1.** Raw data for autoradiography and immunoblotting in *Figure 3B*.

**Figure supplement 1.** Multiple immune signaling branches are partially restored in EFR^F761H/Y836F and EFR^F761H/SSAA.

**Figure supplement 1—source data 1.** Raw data for immunoblots shown in *Figure 3—figure supplement 1C*.

## EFR^F761H/Y836F and EFR^F761H/SSAA restore BIK1 trans-phosphorylation

As shown *Figure 3* and *Figure 1—figure supplement 1*, EFR^Y836F and EFR^SSAA are impaired in elf18-triggered immune signaling at the level of receptor complex activation. In both cases BAK1 S612 phosphorylation is reduced, resulting in compromised ability to *trans*-phosphorylate BIK1. We therefore asked next whether EFR^F761H/Y836F and EFR^F761H/SSAA show restored ability to induce BAK1 autophosphorylation and resulting BIK1 trans-phosphorylation. We performed semi-in vivo IP-kinase assays in

which EFR-GFP variants were immunoprecipitated from 2-week-old seedlings after 10 min mock or 100 nM elf18 treatment, and kinase activity of the complex was assessed by monitoring phosphorylation of recombinant His-BIK1$^{D202N}$ substrate. The EFR$^{WT}$-GFP-BAK1 complex showed pronounced BIK1 phosphorylation compared with the unliganded control (*Figure 3B and C*). In contrast, although BAK1 co-precipitated with both EFR$^{Y836F}$-GFP and EFR$^{SSAA}$-GFP after elf18 treatment, these complexes failed to increase BIK1$^{D202N}$ trans-phosphorylation compared with the unliganded controls. They also showed reduced levels of BAK1 phosphorylation at S612 (*Figure 3B and C*). However, BAK1 S612 phosphorylation (for EFR$^{F761H/Y836F}$ in two out of four experiments) and BIK1$^{D202N}$ trans-phosphorylation were partially restored for EFR$^{F761H/Y836F}$ and EFR$^{F761H/SSAA}$ in this assay (*Figure 3B and C*), arguing that the functional rescue observed in vivo reflects BAK1 activation effects.

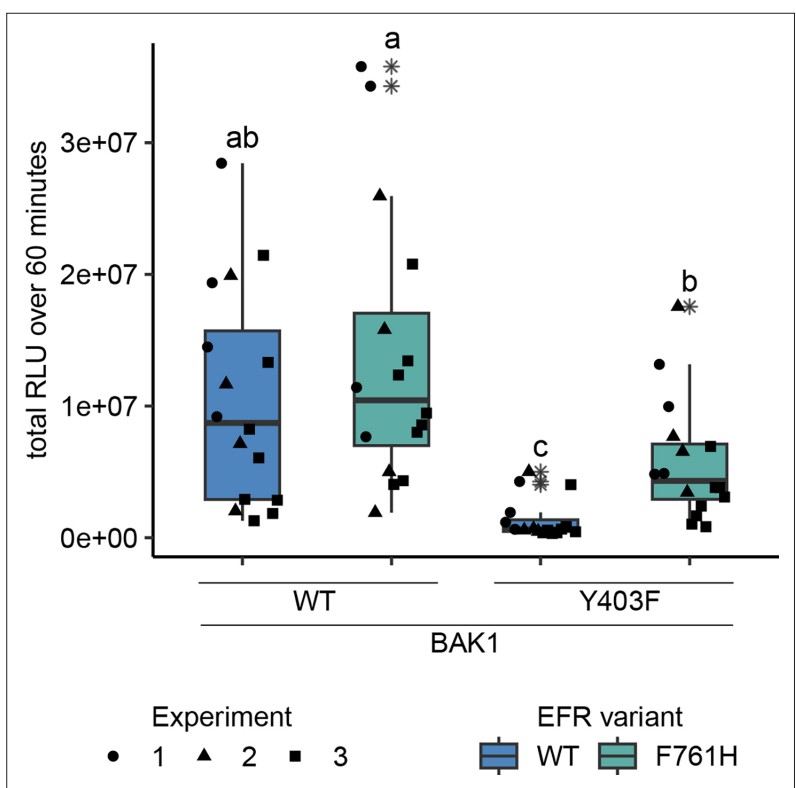

**Figure 4.** EFR$^{F761H}$ recovers BAK1$^{Y403F}$ function. The cytoplasmic domains of BAK1 and EFR variants with fused RiD-tags were transiently expressed in *N. benthamiana* and leaf discs were treated with Rap to induce dimerization. EFR and EFR$^{F761H}$ induced a similar total oxidative burst when BAK1 was co-expressed. The co-expression of BAK1$^{Y403F}$ and EFR diminished the oxidative burst, which was restored partially when EFR$^{F761H}$ was co-expressed. Outliers are indicated by an additional asterisk and included in statistical analysis. Statistical test: Kruskal-Wallis test (p<8.516 *10$^{-7}$), Dunn's post-hoc test with Benjamin-Hochberg correction (p ≤ 0.05) Groups with like letter designations are not statistically different.

The online version of this article includes the following source data and figure supplement(s) for figure 4:

**Figure supplement 1.** Function of BAK1 Y403F is partially recovered by the secondary mutation I338H.

**Figure supplement 1—source data 1.** Raw data for immunoblots shown in *Figure 4—figure supplement 1B*.

**Figure supplement 2.** EFR F761H accelerates the onset of the oxidative burst but requires the catalytic activity of BAK1.

**Figure supplement 2—source data 1.** Raw data for immunoblots shown in *Figure 4—figure supplement 2D*.

**Figure supplement 3.** Protein accumulation for the oxidative burst assay in *Figure 4*.

**Figure supplement 3—source data 1.** Raw data for immunoblots shown in *Figure 4—figure supplement 3*.

## Toward a mechanistic understanding of BAK1 allosteric activation by EFR

A phospho-ablative mutation of the BAK1 VIa-Tyr (Y403F) – analogous to Y836F in EFR – has also been reported to compromise elf18-induced signaling by interfering with receptor complex activation (*Perraki et al., 2018*). We recapitulated this finding using only the intracellular domains of EFR and BAK1, fusing them to FKBP and FRB domains with myristoylation sequences and inducing dimerization with Rap in *N. benthamiana* (*Figure 4* and *Figure 4—figure supplement 1A*). This approach allowed us to investigate BAK1 variants without interference of endogenous NbSERKs. We wondered whether BAK1$^{Y403F}$ could be rescued by a mutation analogous to EFR$^{F761H/M}$ to recover accessibility of the active conformation. Indeed, introducing the analogous mutation (I338H) into BAK1 partly restored the oxidative burst for BAK1$^{I338H/Y403F}$ when dimerized with EFR$^{WT}$ (*Figure 4—figure supplement 1A*), suggesting that BAK1$^{Y403F}$ is perturbed, like EFR$^{Y836F}$, in accessing its active conformation. Analogies with allosteric regulation of other kinases and pseudokinases (*Hu et al., 2013*; *Mace and Murphy, 2021*; *Sheetz and Lemmon, 2022*; *Zhang et al., 2006*) led us hypothesize that EFR might stabilize the active-like conformation of BAK1 intermolecularly – helping to position the BAK1 αC-helix (in the EFR-BAK1 complex) to activate immune signaling. If this is correct, EFR that is partly 'locked' in its active-like conformation may be able to rescue (in trans) the function of a signaling-inactive BAK1 mutant with a destabilized αC-helix (as in BAK1$^{Y403F}$). We therefore asked whether EFR$^{F761H}$ could achieve this, since it exhibited the tendency for increased BIK1$^{D202N}$ trans-phosphorylation in IP-kinase assays (*Figure 3B and C*) and for an accelerated oxidative burst in *N. benthamiana* (*Figure 4—figure supplement 2A and B*). Intriguingly, when EFR$^{F761H}$ was paired with BAK1$^{Y403F}$, the Rap-induced oxidative burst was partially restored (*Figure 4*), suggesting that EFR$^{F761H}$ can partially restore BAK1$^{Y403F}$ function. The fact that no oxidative burst was seen when EFR$^{F761H}$ was paired with catalytically inactive BAK1$^{D416N}$ suggests that this restoration does not occur through a direct catalytic mechanism mediated by EFR$^{F761H}$ (*Figure 4—figure supplement 2C*), supporting the hypothesis of allosteric regulation.

## Catalytic independence is only observed for *Arabidopsis* sp. LRR-RK XIIa kinase domains

EFR belongs to LRR-RK subfamily XIIa (*Figure 5A*), of which two other RKs are functionally described immune RKs in *Arabidopsis*: FLS2 and XPS1 (XANTHINE/URACIL PERMEASE SENSING 1) (*Gómez-Gómez and Boller, 2000*; *Mott et al., 2016*). This subfamily has been implicated more generally in immune signaling, which is supported by induction of immune signaling of in vivo dimerized FLS2, EFR, or XPS1-LIKE 1 (FEXL1) intracellular domains with BAK1 (*Kim et al., 2021*). However, the ligands are not known for most subfamily-XIIa RKs (*Dufayard et al., 2017*; *Kim et al., 2021*). Because EFR does not require its catalytic activity, we wondered whether other LRR-RK XIIa kinase domains similarly function non-catalytically, in a manner similar to pseudokinases. To test this hypothesis, we fused the EFR ectodomain to the transmembrane helix and intracellular domain of different LRR-RK XIIa members to generate elf18 responsive RKs that can dimerize with BAK1/SERKs (*Rhodes et al., 2021*). These chimeric proteins were expressed transiently in *N. benthamiana* leaves, and their immune signaling function was tested by elf18 treatment in oxidative burst assays. All chimeras induced an oxidative burst (*Figure 5B*), except XIIa2 (the closest FLS2-related kinase), which exhibited only a very minor response. To determine whether their signaling function required kinase activity, we next tested variants with mutations in the catalytic site (replacing the putative catalytic base HRD-aspartate with asparagine). EFR$^{D849N}$ induced a robust oxidative burst as expected as well as FEXL1$^{D838N}$ and the closely related XIIa5$^{D839N}$ (AT3G47570). Similarly, XPS1$^{D856N}$ and XIIa6$^{D840N}$ (AT3G47090) induced an oxidative burst independently of their catalytic activity, but the total oxidative burst was reduced compared to the catalytically active variants. Protein levels of catalytic site mutants accumulated comparably to their corresponding wild-type versions (*Figure 5—figure supplement 2*). In contrast, the catalytically inactive FLS2$^{D997N}$ showed a diminished oxidative burst (*Figure 5B*), suggesting that it may be mechanistically distinct. Furthermore, kinetic differences for the oxidative burst of *Arabidopsis* XIIa kinase domains were observed. XPS1$^{D856N}$ and XIIa6$^{D840N}$ showed a delayed oxidative burst (*Figure 5—figure supplement 1A and B*). Also, EFR$^{D849N}$ exhibited a delay of approximately 5–7 min compared to wild-type EFR. In contrast, oxidative bursts induced by FEXL1$^{D838N}$ and XIIa5$^{D839N}$ were only slightly delayed compared to their corresponding catalytically active kinase domains. We corroborated these findings for XIIa5 by performing in vitro kinase assays, observing that BIK1$^{D202N}$

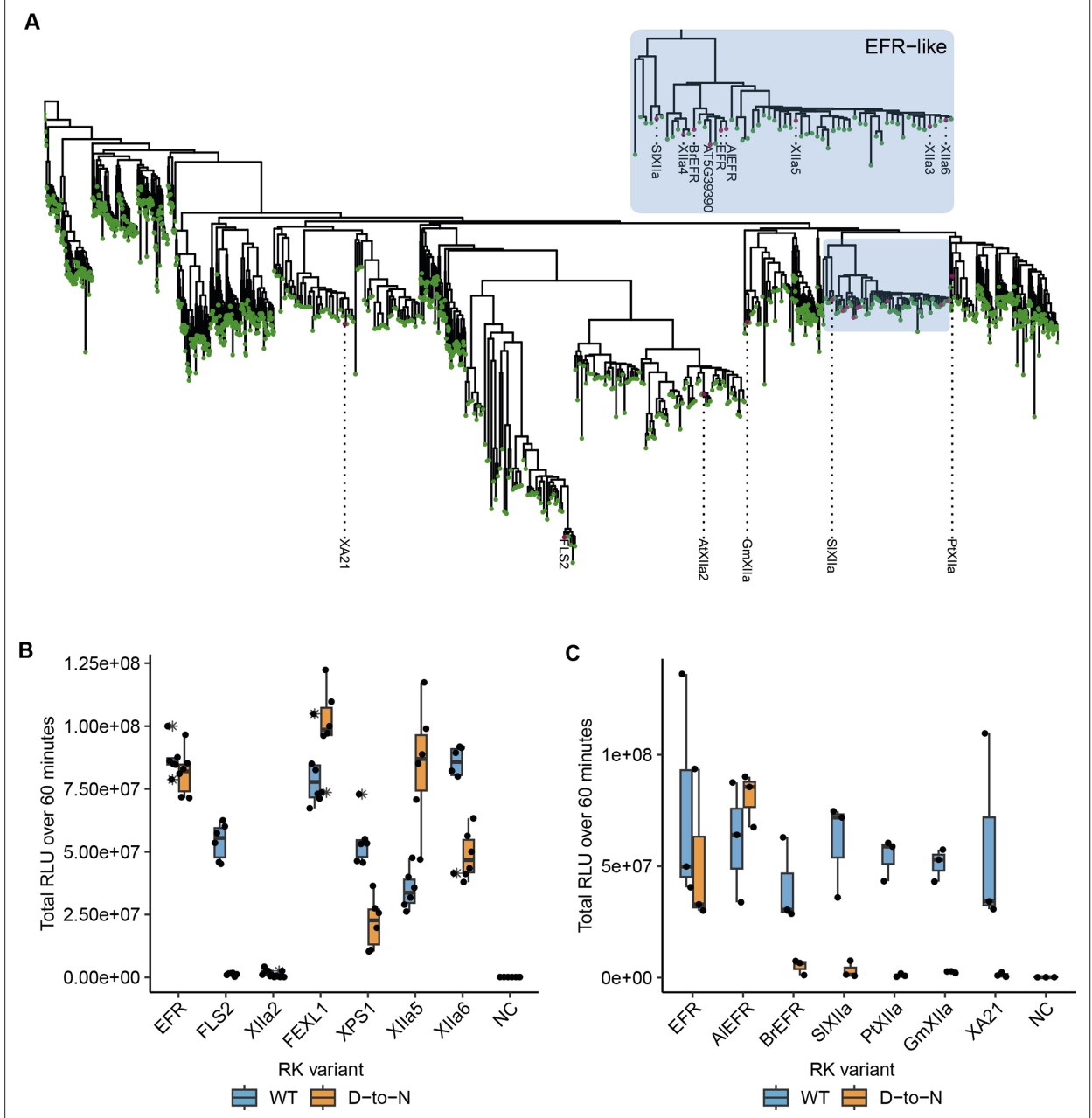

**Figure 5.** Related EFR kinases from LRR-RK XIIa in the *Arabidopsis* genus can function independent of their calatytic activity. (**A**) Phylogenetic analysis of LRR-RK subfamily XIIa. Selected LRR-RK XIIa kinase domains are labeled and highlighted with purple points. The EFR-like clade contains all *Arabidopsis* XIIa kinases except FLS2 and XIIa2 and also selected XIIa kinases from *Arabidopsis lyrata* and *Brassica rapa*. (**B, C**) The ectodomain of EFR was fused to the transmembrane and intracellular domain of selected LRR-RK XIIa members to create elf18-responsive chimeras for testing the immune signaling function and catalytic dependency of the related kinase domains. The chimeras were transiently expressed in *N. benthamiana* and tested in oxidative burst assays. All *Arabidopsis* LRR-RK XIIa members induced an oxidative burst except XIIa2, the closest FLS2 related kinase in the subfamily. Catalytic dependency of the kinase domains appears to vary from kinase to kinase, with catalytically dead versions of EFR, FEXL1 and XIIa5 inducing a WT-like oxidative burst and XPS1 and XIIa6 displaying a reduced oxidative burst. FLS2 kinase dead exhibited a diminished oxidative burst. Experiments were repeated three times with similar results.

The online version of this article includes the following source data and figure supplement(s) for figure 5:

**Figure supplement 1.** XIIa5D839N exhibits largely XIIa5WT-like characteristics.

**Figure supplement 1—source data 1.** Raw data for autoradiography shown in *Figure 5—figure supplement 1C*.

**Figure supplement 2.** Protein accumulation of EFR-XIIa chimeras in *N. benthamiana*.

**Figure supplement 2—source data 1.** Raw data for autoradiography shown in *Figure 5—figure supplement 2*.

and BAK1 phosphorylation increased after Rap application as effectively for XIIa5[D839N] as for wild-type XIIa5 (*Figure 5—figure supplement 1C and D*). Taken together, our results imply that catalytic activity for XIIa5 is almost fully dispensable.

Because most *Arabidopsis* LRR-RK XIIa kinase domains execute their signaling function at least to some extent independently of their catalytic activity, we went on to test XIIa kinase domains from other species, that we selected from a phylogenetic tree of LRR-RK XIIa kinases built from previous phylogenetic analysis (*Figure 5A*, *Dufayard et al., 2017*). The selected catalytically active XIIa kinase domains from *Oryza sativa* (XA21), *Populus trichocarpa* (PtXIIa), *Solanum lycopersicum* (SlXIIa), *Glycine max* (GmXIIa), *Brassica rapa* (BrEFR), and *Arabidopsis lyrata* (AlEFR), induced an oxidative burst (*Figure 5C*; *Figure 5—figure supplement 1A and B*). However, catalytic site mutants of all non-*Arabidopsis* XIIa RKs accumulated similar to their wild-type counterparts and abolished their function in oxidative burst assays (*Figure 5*, *Figure 5—figure supplement 2*).

## Discussion

Our study provides insights into the non-catalytic activation mechanisms of the EFR-BAK1 complex. Rather than requiring its kinase activity to transactivate BAK1, our results suggest that the EFR kinase domain undergoes a 'conformational toggle' to an active-like state that permits allosteric *trans*-activation of BAK1. EFR function is impaired by perturbation of structural elements that are important for conformational switching – specifically, mutation of the subdomain VIa Y836 or ablation of A-loop phosphorylation sites – but these effects can be reversed by intragenic suppressor mutations that stabilize the active-like conformation, notably the F761H mutation.

Non-catalytic functions of kinases and pseudokinases are human disease-relevant and have emerged as important signaling regulators in metazoans (*Eyers and Murphy, 2013*; *Mace and Murphy, 2021*; *Schmidt et al., 2021*; *Sheetz and Lemmon, 2022*). By comparison, very little is known about pseudokinases or non-catalytic kinase functions in plants, although pseudokinases are as prevalent in plant as in metazoan kinomes (*Kwon et al., 2019*). Our data demonstrate that EFR and potentially other *Arabidopsis* XIIa kinases function as allosteric regulators of BAK1 kinase activity, implying pseudokinase-like functions. Thus, our findings establish precedence for non-catalytic mechanisms, specifically allosteric regulation, in plant RK signaling.

Phosphorylation of EFR is crucial for signaling function and potentially supports the conformational toggle. Nevertheless, our results do not resolve how phosphorylation of the A-loop or VIa-Tyr facilitate conformational toggling for EFR, and structural studies will be required to address this question. However, our HDX-MS analysis of unphosphorylated protein suggests that EFR[Y836F] causes the active-like conformation to become inaccessible, potentially due to a lack of an inter-lobe H-bond – consistent with earlier studies describing the corresponding VIa-Tyr as a pivot for αC-helix movements in PKA (*Taylor and Kornev, 2011*; *Tsigelny et al., 1999*). Recent analysis of the PKA hinge region demonstrates the importance of its integrity for coordinated conformational changes in the N- and C-lobe (*Olivieri et al., 2023*; *Wu et al., 2023*). Whether EFR VIa-Tyr phosphorylation contributes to these coordinated conformational changes is unclear. We attempted to directly address the role of pY836 in EFR conformational dynamics using HDX-MS but were unable to produce sufficient recombinant protein with a pTyr-analog incorporated at the Y836 site. However, phosphorylation of the VIa-Tyr presumably distorts the hinge region due to its proximity to the αC-β4-loop. Alternatively, the VIa-pTyr and C-terminal end of the αE-helix must rotate outwards (*Figure 2—figure supplement 1*). Rotation of the αE-helix, however, would have direct impact on the positioning of the catalytic loop and RS1. Of note, functional importance of VIa-Tyr phosphorylation was assigned based on correlation between its phosphorylation and functional impairment by phenylalanine substitution in multiple plant RKs (*Liu et al., 2018*; *Luo et al., 2020*; *Macho et al., 2014*; *Perraki et al., 2018*), making it possible that functional importance of its phosphorylation confounded a structural role. Future computational protein-structure prediction techniques that consider post-translational modifications will aid in elucidating the effect of EFR VIa-Tyr phosphorylation on EFR's conformation. Nevertheless, mutation of VIa-Tyr may be a useful tool to break pseudokinase/non-catalytic functions by impeding conformational toggling in a way that is more effective than using conventional kinase-dead mutations.

Basal BAK1 activity in vitro is low compared to its robustly enhanced activity after Rap-induced dimerization (*Figure 1B and C*; *Figure 5—figure supplement 1C and D*), which might be further reduced in planta by negative regulators (e.g. protein phosphatases; *Segonzac et al., 2014*). We

propose that, by increasing the local concentration, ligand-induced association of EFR and BAK1 allows partially active BAK1 to phosphorylate the adjacent EFR A-loop. This in turn allows wild-type EFR to adopt the active-like conformation that can allosterically fully activate BAK1 to promote substrate phosphorylation. The activating mutation EFR F761H potentially circumvents the requirement of A-loop phosphorylation since it partially restored EFR$^{SSAA}$ function. Consistently, EFR$^{F761H}$ showed elevated BIK1 trans-phosphorylation in IP-kinase assays and an accelerated oxidative burst (*Figure 3B and C*; *Figure 4—figure supplement 2A, B*), suggesting constitutive assembly of the active-like conformation. However, EFR$^{F761H}$ still requires catalytic activity of complexed BAK1 (*Figure 4—figure supplement 1*).

Our data demonstrate that allosteric BAK1 kinase activation plays a key role in EFR-BAK1-mediated immune signaling. The partial recovery of BAK1$^{Y403F}$, which we hypothesize is impaired in αC-helix positioning, by EFR$^{F761H}$ suggests a mechanism for allosteric regulation involving BAK1 αC-helix positioning. Indeed, multiple metazoan kinases are allosterically regulated by αC-helix positioning (*Sheetz and Lemmon, 2022*). Moreover, BAK1 has a high propensity for a disordered αC-helix (*Moffett et al., 2017*), suggesting its correct positioning requires additional support. Different orientations of kinase-kinase dimers, in which allosteric regulation occurs at the αC-helix, were previously described (*Mace and Murphy, 2021*; *Sheetz and Lemmon, 2022*). Structural analysis will be required to resolve the exact interaction interface for allosteric activation in the EFR-BAK1 kinase dimer and to design interface-disrupting mutations for structure-function analysis but is beyond the scope of the present work.

Allosteric activation of BAK1/SERKs is partially preserved in *Arabidopsis* LRR-RK XIIa kinases (*Figure 5*). Whether these kinases function solely non-catalytically, as suggested by XIIa5$^{D839N}$ (*Figure 5*, *Figure 5—figure supplement 1*), or contribute also catalytically to signaling activation is unclear since oxidative bursts induced by all functional catalytic base mutants of XIIa kinases were delayed (*Figure 5—figure supplement 1A and B*). Catalytic base mutation may affect conformational dynamics, which we show are important for EFR function (*Figure 2*), confounding our interpretation. Inactive XIIa kinases with intact conformational dynamics may clarify the extent of catalytic contribution and could be obtained by development of selective inhibitors or engineering analog sensitive kinases. Alternatively, catalytic independency could be a recent innovation in *Arabidopsis* sp. XIIa kinases, that is taxonomically restricted due to a strong negative selection on the kinase domain (*Man et al., 2023*) but further study is required to test this hypothesis.

Taken together, our results add to growing evidence that non-catalytic functions of kinases are similar to *bona fide* kinases controlled by conformational switching (*Sheetz et al., 2020*; *Sheetz and Lemmon, 2022*) and further set precedence for discovering more non-catalytic mechanisms in plant RK signaling where pseudokinases are particularly prevalent (*Kwon et al., 2019*).

## Limitations of the study

In our in vitro kinase assays using the RiD system, the proteins were freely diffusing as they were produced as soluble entities. Freely diffusing components, however, do not fully resemble the in planta mode of activation where the plasma membrane restricts the free movement. The colocalization of RK/coRK complexes with their substrates may be a supporting component of signaling activation, which we did not capture in our in vitro assays.

It is yet unclear whether EFR phosphorylates BAK1 efficiently on activation-relevant residues in the context of stabilized proximity. This information will ultimately help to deduce whether the catalytic activity of EFR contributes to signaling activation and whether even marginal catalytic activity of EFR$^{D849N}$ may be sufficient to support signaling activation.

Further, our study relies heavily on mutations. Mutations can have pleiotropic effects as they impact conformational plasticity which supports catalytic activity and non-catalytic functions. Thus, it is difficult to untangle catalytic from non-catalytic functions using solely mutations. Eventually, it would be beneficial to obtain structural information of native proteins in a complex and monitor site-specific and time-resolved phosphorylation on complex components.

We do not have experimental support for EFR$^{F761H}$ stabilizing or facilitating the transition into an active-like conformation and relieving the negative effect of the Y836F mutation in EFR$^{F761H/Y836F}$. It would have been beneficial to obtain HDX-MS data for these mutants as well, but we could not produce sufficient protein to do so. Thus, we rather derived the biochemical and structural effects of the EFR$^{F761H}$ mutant by homology to mammalian kinases, for example BRAF.

## Materials and methods

### Plant material and growth conditions

For complementation experiments, the *efr-1* T-DNA insertional mutant was used (*Zipfel et al., 2006*). A comprehensive list of transgenic line used in this study can be found in the Key Resources Table.

For sterile plant culture, the growth conditions were 120 µmol s$^{-1}$ m$^{-2}$ illumination during 16 hr light/8 hr dark cycles at a constant temperature of 22 °C. Sterilization of seeds was performed by chlorine gas surface sterilization for 6 hr. Sterile seeds were germinated on 0.8% (w/v) phyto agar plates containing 0.5 x Murashige and Skoog (MS, Duchefa) basal salt mixture and 1% (w/v) sucrose. After 5 days of growth on agar plates, seedlings were transferred to liquid 0.5 x MS medium containing 1% (w/v) sucrose in either sterile 6- (IP-kinase), 24- (MAPK activation), or 48-well (seedling growth inhibition) plates.

For plant growth on soil, seeds were resuspended in 0.05% (w/v) agarose solution and stratified for at least 16 hr in the dark at 4 °C. Seeds were then directly sown on soil using a Pasteur pipette.

### Physiological assays

#### Seedling growth inhibition assay

Upon transfer to liquid culture, seedlings were exposed to either mock (no PAMP supplementation) or PAMP (5 nM elf18) treatment. After 10 days of seedling growth in liquid culture, seedlings were dry blotted on a paper towel to remove excess liquid media before measuring their fresh weight using a fine balance (Sartorius X64-S1). The relative seedling weight was calculated by dividing the seedling weight of each PAMP treated seedling by the average seedling weight of all mock-treated seedlings of the respective genotype.

#### MAPK activation assay

After transfer to liquid media, seedling growth continued for 10 days. The liquid medium was then decanted from the 24-well plate, and 1 ml of liquid 0.5 x MS-medium containing either no elf18 supplementation (mock) or 100 nM elf18 supplementation was added to each well and seedlings incubated for 10 min. Thirty seconds prior to the end of this incubation, seedlings were dry blotted on a paper towel, transferred to a 1.5 ml reaction tube, and flash frozen in liquid nitrogen. Frozen tissue was then stored at –80 °C.

For protein extraction, the frozen tissue was ground using a plastic pestle in the 1.5 ml microcentrifuge tube and analyzed as described.

#### Measurement of apoplastic oxidative burst

Four- to 5-week-old under a short day regime (130 µmol s$^{-1}$ m$^{-2}$, 65% humidity, 10 hr/14 hr light/dark-cycle) soil-grown plants were used for punching out at least two leaf discs (4 mm in diameter) per plant. Leaf discs were floated on ultrapure water in a white chimney 96-well plate. The next day, the water solution was replaced with assay solution containing 100 nM elf18, 100 µM luminol and 10 µg/ml horseradish peroxidase, immediately before recording luminescence in 1 min intervals with 250 ms integration time per well in a Tecan Spark plate reader. The luminescence recorded of all leaf discs coming from the same plant were averaged for each time point. For plotting the time course of recorded luminescence emission, the means of all plants belonging to the same genotype were averaged and the standard error of the mean was calculated, which is represented by the error bar. For calculating the time to half maximum, the timepoint of maximum oxidative burst was determined and the next five values included for fitting a sigmoidal curve to the oxidative burst of each leaf disc. From the fitted function, the half maximum was derived and all half maxima from leaf discs belonging to one plant were averaged.

#### Infection assay with GUS activity monitoring

Three to four-week-old plants were infiltrated with *A. tumefaciens* carrying pBIN19-GUS(intronic) at OD$_{600}$=0.5. *A. tumefaciens* was inoculated the day before and grown overnight. Five days after infiltration, infiltrated leaves were harvested into a 2 ml microcentrifuge tube with two ø4 mm glass beads and flash frozen in liquid nitrogen. Plant tissue was ground in a GenoGrinder (90 s, 1500 rpm). A total of 600 µl of extraction buffer (50 mM NaH$_2$PO$_4$-NaOH, pH 7.0, 10 mM EDTA, 0.1% (v/v) Triton X-100,

0.1% (v/v) sodium lauroyl sarcosinate, 10 mM β-mercaptoethanol) was added to tissue powder and incubated for 30 min on a rotator at 4 °C. Protein extracts were centrifuged (10 min, 13,000 rpm, 4 °C) in a table top centrifuge, and 200 µl supernatant was collected. 100 µl of the supernatant was mixed with 100 µl extraction buffer containing 2 mM 4-methylumbelliferyl-β-D-glucopyranoside and incubated for 30 min at 37 °C. 40 µl of the reaction was collected and mixed with 160 µl $Na_2CO_3$ to stop the reaction and enhance the fluorescence of 4-methylumbelliferone. A standard curve of 4-methylumbelliferone in extraction buffer was prepared, starting with 10 µM and using twofold dilution steps. Finally, fluorescence was measured in a plate reader. Protein concentration was determined using Bradford reagent with 1:20 dilution of protein extract. Fluorescence was converted into 4-MU concentration using the standard curve and 4-MU concentration was normalized to the amount of protein in the 40 µl sample divided by the incubation time.

## Molecular cloning

All primers and plasmids used and generated in this study are listed in the Key Resources Table.

For recombinant expression and in planta complementation, the gene's cDNA sequences were subcloned from previously published cDNA clones (*Bender et al., 2021*; *Perraki et al., 2018*) or synthesized with domesticated BsaI, BpiI and Esp3I sites. PCR products were inserted into level 0 Golden-Gate plasmids pICSL01005, pICH41308 (*Weber et al., 2011*), or the universal acceptor p641 (*Chiasson et al., 2019*) respectively. GoldenGate reactions were performed with 5 U of restriction enzyme and 200 U of T4 ligase in T4 ligase buffer (NEB) also containing 0.1 mg/ml BSA (NEB; *Weber et al., 2011*). GoldenGate digestion ligation cycles varied between 10 and 20.

Site directed mutagenesis (SDM) was conducted as described (*Liu and Naismith, 2008*) or during GoldenGate cloning by amplifying the target in two pieces that were ligated in the restriction-ligation reaction with the intended nucleotide changes in the restriction overhangs. Where SDM was performed according to *Liu and Naismith, 2008*, the PCR reaction was DpnI (New England Biolabs) digested (37 °C, 1–2 hr) without prior clean-ups, and then transformed into *E. coli* DH10b.

### Construction of pICSL86955-35S::EFRecto-ccdB-mEGFP::HSP18t

The ccdB counter selection cassette was amplified from p641-Esp3I, and the EFR ectodomain sequence was amplified from pICSL01005-EFR using primers listed in the Key Resources Table. The PCR products were purified and used together with level 0 GoldenGate plasmids for 35 S promoter +TMVOmega 5'UTR, C-terminal mEGFP-tag and HSP18 terminator for assembly into pICSL86955 using BsaI (ThermoScientific) restriction enzyme and T4 DNA ligase (New England Biolabs). The GoldenGate restriction-ligation-reaction was transformed into *E. coli* One Shot ccdB Survival 2 T1R Competent Cells (Thermo Fisher). Single colonies were cultured for plasmid isolation and the cloned sequence was confirmed by DNA sequencing.

### Construction of p641-BsaI

Site-directed mutagenesis was performed according to *Liu and Naismith, 2008*, using primers listed in the Key Resources Table using p641-EspI as template.

### Construction of pETGG

The plasmid pET28a(+)–6xHis-TEV-GB1 was linearized by PCR excluding 6xHis-TEV-GB1 sequences and the C-terminal 6xHis-tag that were replaced by the GoldenGate cloning cassette containing AATG and GCTT BsaI restriction sites, the ccdB counter selection marker, and the chloramphenicol resistance gene. The GoldenGate cloning cassette was amplified using primers listed in the Key Resources Table using p641-BsaI as template and the PCR product was ligated with the linearized pET28a(+) backbone using InFusion (Takara) cloning.

## Plant transformation

### Stable transformation of *Arabidopsis thaliana*

For complementation of *efr-1* T-DNA knock-out lines, the coding sequence of EFR and the mutants were cloned in the GoldenGate system as described in the section *Molecular cloning*. Sequences of final binary plasmids were confirmed by sequencing prior to transformation into *Agrobacterium*

*tumefaciens* GV3101 by electroporation (25 µF, 200 Ω, 1.8 kV). Plants were grown to the early flowering stage and then transformed using the floral dip method with bacteria grown in YEBS medium (*Clough and Bent, 1998*; *Davis et al., 2009*). Seeds after dip transformation were selected on 0.5 x MS-agar plates containing 10 µg/ml phosphinothricin until homozygous seed batches were identified in T3 generation. These seed batches were then used for physiological assays.

## Transient transformation of *Nicotiana benthamiana*

*Agrobacterium tumefaciens* was grown over night in liquid LB medium supplemented with kanamycin (50 µg/ml), gentamycin (25 µg/ml) and rifampicin (40 µg/ml). Cultures were diluted the next morning 1:10 in fresh LB medium (without antibiotics) and grown to an optical density $OD_{600}$ 0.8–1.2. Cultures were spun down (2000 rcf, 10 min) and LB medium was decanted. Pellet was resuspended in infiltration medium (10 mM MES-KOH, pH 5.8, 10 mM $MgCl_2$), the optical density ($OD_{600}$) was determined in a spectrophotometer. For infiltration, *A. tumefaciens* carrying the construct with the gene of interest and a second strain carrying the RNA silencing suppressor p19 were mixed at a 2:1 ratio (final $OD_{600}$=0.5 + 0.25=0.75). The mixture was infiltrated into 4- to 5-week-old *N. benthamiana* plants from the abaxial site of the leaf with a needle-less 1 ml syringe. Constructs within one experiment were infiltrated side-by-side into the same leaf on multiple plants. Only for testing the catalytic requirement of multiple LRR-RK XIIa kinase domains, not all constructs could be infiltrated side-by-side on one leaf.

## Protein extraction from plant samples

Flash frozen tissue was ground using plastic pestles in 1.5 ml microcentrifuge tubes or, in case of coIP and IP-kinase samples, using stainless steel grinding jars and a Retsch mill (90 s, 30 Hz). Ground tissue was mixed with extraction buffer 50 mM Tris-HCl, pH 7.5, 150 mM NaCl, 10% glycerol, 2 mM EDTA, 1% IGEPAL detergent, 1 mM DTT, 4 mM sodium tartrate ($Na_2C_4H_4O_6$), 1% (v/v) protease inhibitor cocktail (P9599, Sigma), 1 mM PMSF, 2 mM sodium molybdate ($Na_2MoO_4$), 1 mM sodium fluoride (NaF), and 1 mM activated sodium orthovanadate ($Na_3VO_4$) at an 1:1 – 1:2 ratio (tissue powder:extraction buffer) and incubated for 30–45 min on a rotator at 4 °C. For, coIP or IP-kinase assays, the samples were filtered through two layers of Miracloth into conical centrifugation tubes, which were spun at 20,000 rcf for 20 min, and supernatant was collected. Protein concentrations in the supernatants were determined using Bradford reagent. Subsequently, protein concentration was adjusted to normalize samples.

## Recombinant protein expression and purification

For HDX-MS pET28a(+)–6xHis-EFR and pET28a(+)–6xHis-EFR Y836F were transformed by heat shock into BL21(DE3) pLPP (Amid Biosciences). Protein expression and purification was performed as described for proteins produced for the in vitro kinase assay. Conditions for growth and extraction are detailed in *Table 2*, except, that protein was eluted from the gel filtration column in 20 mM HEPES, pH 7.2 (NaOH), 150 mM NaCl.

Unphosphorylated protein for in vitro RiD kinase assays were recombinantly produced in *E. coli* BL21(DE3)-V2R-pACYC LamP (*Wernimont et al., 2010*). From a single colony, a 10 ml lysogeny broth (LB) starter culture supplemented with 50 µg/ml kanamycin and 15 µg/ml chloramphenicol

**Table 2.** Protein expression conditions.

| Protein | T after induction [°C] | t of expression [h] | pH of extraction buffer |
|---|---|---|---|
| For in vitro kinase assay | | | |
| EFR/EFR D849N | 30 | 4 | 8.0 |
| BAK1 | 30 | 4 | 8.0 |
| BRI1/BRI1 D1009N | 18 | Over night | 8.0 |
| FLS2/FLS2 D997N | 18 | Over night | 8.0 |
| BIK1 D202N | 18 | Over night | 7.5 |
| For HDX-MS | | | |
| EFR/EFR Y836F | 30 | 3 | 8.0 |

was inoculated and incubated overnight at 37 °C, shaking at 220 rpm. On the next day, the starter culture was completely transferred to 1 l of LB medium containing 50 µg/ml kanamycin at 37 °C with shaking to an $OD_{600}$ of 0.6–0.8. With the addition of 300 µM isopropyl β-D-1-thiogalactopyranoside (IPTG), expression of recombinant protein was induced, and growth continued with conditions indicated in *Table 2*. Cells were then pelleted by centrifugation at 4000 rcf for 10 min, and pellets resuspended in protein extraction buffer (20 mM HEPES-NaOH at varying pH (see *Table 2*), 500 mM NaCl, 10 mM imidazole, 5% glycerol). Resuspended bacterial pellets were stored at –80 °C before protein purification.

Frozen pellet suspensions were thawed in a water bath at room temperature and then transferred to ice. Cells were lysed using ultrasonication (Branson Sonifier 250) with a ø6 mm sonicator probe at 60% amplitude, with 20 s ON/40 s OFF intervals for a total of 4 cycles. Cell debris was pelleted by centrifugation at 47,850 rcf for 30 min at 4 °C. The supernatant was collected and equilibrated Pure-Cube 100 Co-NTA agarose beads (Cube Biotech) were added for batch-binding of protein. Binding continued for 45 min while rotating at 14 rpm on a tabletop rotator at 4 °C. Beads were collected by centrifugation at 500 rcf for 1 min at 4 °C, and supernatant was removed using a vacuum pump. Beads were then washed twice with 10 ml ice-cold protein extraction buffer, and protein eluted in elution buffer (20 mM HEPES-NaOH, pH 8.0, 300 mM NaCl, 300 mM imidazole, 5% glycerol) by incubation for 10 min, followed by spinning down beads and collecting supernatants. Eluted protein was then filtered through a 0.22 µm spin-column filter (1 min, 4,000 rcf, 4 °C) before loading onto a Superdex 200 Increase 10/300 GL (Cytiva) gel filtration column equilibrated with 20 mM HEPES-NaOH, pH 7.5, 200 mM NaCl, 5% glycerol using an Äkta pure protein purification system (Cytiva). The peak fraction was collected, concentration measured by NanoDrop, and protein aliquoted and snap frozen in liquid nitrogen prior to storage at –80 °C.

## Hydrogen-deuterium exchange and mass spectrometry (HDX-MS)

All HDX data were collected using LEAP HDX automation (Trajan). A 5 µL volume of freshly purified EFR[WT] or EFR[Y836F] at 0.3–0.6 mg/ml in 20 mM HEPES-NaOH, pH 7.2, 150 mM NaCl (5 µl) was labeled by 20-fold dilution with 20 mM HEPES-NaOH, pD 7.4, 100 mM NaCl at 25 °C. The labeled sample was quenched at different time points (10, 60, 600, 3600, and 7200 s) by adding 100 µl of cold 200 mM glycine buffer (pH 2.3). A fully-deuterated sample was also prepared by labeling the protein sample for 1 min with 20 mM HEPES-NaOH, pD 7.4, 100 mM NaCl, 8 M urea-d4 (Cambridge Isotope Laboratories, Inc). The quenched sample was immediately injected onto an Enzymate BEH pepsin column (Waters) at 2 °C, and the labeled sample was digested for 3 min. The peptic peptides were trapped and separated using an Acquity UPLC BEH C18 pre-column (2.1x5 mm, 1.7 µm, Waters) and Acquity UPLC BEH C18 column (1.0x100 mm, 1.7 µm, Waters), respectively, using a linear gradient of 5 to 40% acetonitrile over 7 min. The MS[e] data were acquired on a Synapt G2-Si (Waters) using 0.5 s scan time and ramp collision energy of 5 V to 10 V for LE and 15 V to 40 V for HE with continuous lock mass (Leu-Enk) for the mass accuracy correction.

**Table 3.** Summary table of HDX-MS analysis.

| Data Set | Wild-type EFR | EFR[Y836F] |
| --- | --- | --- |
| HDX reaction details | 20 mM HEPES, 100 mM NaCl, pD 7.4 | 20 mM HEPES, 100 mM NaCl, pD 7.4 |
| HDX time course | 0, 10, 60, 600, 3600, and 7200 s | 0, 10, 60, 600, 3600, and 7200 s |
| HDX control samples | 8 M urea-D4 for fully deuterated standard | 8 M urea-D4 for fully deuterated standard |
| Back-exchange | 46% | 49% |
| # of Peptides | 143 | 153 |
| Sequence coverage | 100% | 100% |
| Ave peptide length / Redundancy | 12.7/5.44 | 13.3/6.06 |
| Replicates | n=3 biological repeats (3 technical/n) | n=3 biological repeats (3 technical/n) |
| Repeatability (Ave SD) | 0.129 Da (2.28 %) | 0.111 Da (2.09 %) |
| Significant differences in HDX (98% CL) | 0.395 Da (7.18 %) | |

**Table 4.** Extinction coefficients and molecular weights retrieved from ProtParam and used for determination of protein concentration.

| Protein | Extinction coefficient (reduced state) [Abs 0.1% (=1 g/l)] | Molecular weight [kDa] |
|---|---|---|
| FKBP-EFR-mEGFP | 7.40 | 80 |
| FKBP-BRI1-mEGFP | 8.26 | 83 |
| FKBP-FLS2-mEGFP | 6.56 | 79 |
| FKBP-XIIa5-mEGFP | 7.59 | 79.5 |
| FRB-BAK1 | 15.07 | 55 |
| 6xHis-TEV-BIK1[D202N] | 9.45 | 47 |
| 6xHis-TEV-EFR(684–1031) | 6.97 | 41.5 |

## HDX-MS data analysis

Peptides were sequenced using ProteinLynx Global Server 3.03 (PLGS, Waters), and the deuterium uptake of each peptic peptides was determined using DynamX 3.0 (Waters). The deuterium uptake of all analyzed peptides presented in this study is the average uptake of three biological replicates with technical triplicates per biological sample. The percent exchange of each peptic peptide (%D) was calculated by the following equation:

$$\%Ex = \left( m_t - m_0 \right) / \left( m_f - m_0 \right) \cdot 100$$

where $m_t$ = the centroid mass of a peptic peptide at time, t, $m_0$=the centroid mass of a peptic peptide without deuterium labeling, and $m_f$ = the centroid mass of a peptic peptide for the fully-deuterated standard sample. The Student's t-test of the HDX data was calculated, as described previously (*Houde et al., 2011*), by using the average of standard deviations of the percent exchange data of all analyzed peptides from n=3 biological experiments. All data were collected and analyzed according to consensus HDX-MS guidelines (*Masson et al., 2019*). A summary of HDX-MS data is presented in *Table 3*; *Table 2*.

## In vitro kinase assay

Aliquots of purified protein stored at –80 °C were thawed in a water bath at room temperature and then kept on ice. Protein concentrations were determined by measuring the absorption at 280 nm using a Nanodrop 1000 Spectrophotometer (ThermoScientific) and calculating the concentration using the computed extinction coefficient under reducing conditions (Expasy Protparam, *Table 4*) of the respective protein. Kinase reactions were performed with 50 nM of each kinase in a total reaction volume of 20 µl containing 20 mM HEPES-NaOH, pH 7.2, 2.5 mM MgCl$_2$, 2.5 mM MnCl$_2$, 1 mM DTT, 100 µM ATP, and 0.5 µCi $^{32}$γP-ATP. A total of 500 nM of 6xHis-TEV-BIK1 D202N was also added to the reaction mixture. The reaction was stopped after 10 min by adding 5 µl 6 x SDS-loading buffer (300 mM Tris, pH6.8, 30% (v/v) glycerol, 6% (w/v) SDS, 0.05% (w/v) bromophenol blue), and heating the sample at 70 °C for 10 min. Subsequently, 20 µl of the sample was loaded onto a 10% SDS-PAGE gel and protein separated by electrophoresis at 130 V for 60–70 min. Proteins were then transferred to a PVDF membrane at a current of 200 mA over 2 hr. Membranes were stained with CBBG250 for 20 s and destaining (45% methanol (v/v), 10% acetic acid (v/v)) for 10 min. Finally, a phosphor-screen was exposed to the PVDF membrane overnight and imaged using an Amersham Typhoon (GE Lifesciences). Band intensities were quantified using ImageQuant software (GE Lifesciences) with background subtraction using the local median method.

## IP-kinase assay/co-immunoprecipitation

Forty microliters of a 50%-slurry of GFP-Trap agarose-beads (ChromoTek) per sample were equilibrated, first with 1 ml water and then twice with 1 ml extraction buffer. Beads were then prepared as a 50% slurry in extraction buffer, and 40 µl were added to each protein extract prepared as described above. Beads were incubated with the extract for 2 hr at 4 °C on a rotator. Beads were then washed four times with 1 ml extraction buffer and split into halves at the last washing step. To one half, 20 µl

2 x SDS-loading buffer was added, and the sample heated for 5 min at 95 °C. The other half was equilibrated with 500 µl kinase reaction buffer (20 mM HEPES, pH 7.2, 5% glycerol, 100 mM NaCl). After pelleting the beads, the supernatant was aspirated and 20 µl of kinase reaction buffer containing additionally 2.5 mM MgCl$_2$, 2.5 mM MnCl$_2$, 100 µM ATP, 1 µCi $^{32}$γP-ATP and 0.5 µM 6xHis-BIK1 D202N were added to each sample. Kinase reactions were incubated at 30 °C for 30 min with 800 rpm shaking and were stopped by adding 5 µl 6 x SDS-loading dye and heating at 70 °C for 10 min. Subsequent steps were performed as described for in vitro kinase assays.

## SDS-PAGE and western blotting

SDS containing gels were prepared manually. The resolving gel buffer contained 0.375 M Tris base, 0.4% SDS, pH 8.8, 10–12% acrylamide (37.5:1 acrylamide:bisacrylamide ratio), and the stacking gel buffer contained 0.125 M Tris base, 0.4% SDS, pH 6.8, 5% acrylamide (37.5:1 acrlyamid:bisacrylamide ratio). Polymerization was induced by addition of 1 mg/ml ammonium persulfate and 1:2000 TEMED in case of the resolving gel or 1:1000 TEMED for the stacking gel.

Protein samples were mixed with 6 x SDS loading dye (300 mM Tris, pH 6.8, 30% glycerol, 6% SDS, 0.05% bromophenol blue) and DTT was added to a final concentration of 100 mM. Samples were heated to 80–90°C for 5–10 min prior to loading the gel. Electrophoresis, with gels being submerged in SDS running buffer (25 mM Tris, 192 mM glycine, 0.1% SDS) was performed at 120–200 V until the dye front reached the bottom of the gel.

Subsequently, proteins were transferred onto PVDF membranes using wet transfer. For this, the transfer stack was assembled fully submerged in transfer buffer (25 mM Tris base, 192 mM glycine, 20% MeOH). Transfer was performed at 100 V for 90 min in the cold room (4 °C) with an additional ice pack in the casket. Membranes were subsequently blocked with 5% skim milk powder dissolved in Tris buffered saline containing Tween-20 (TBS-T; 20 mM Tris base, pH 7.5, 150 mM NaCl, 0.1% Tween-20) for at least 2 hr. The primary antibody was then added (refer to Key Resources Table for conditions) and binding allowed overnight on a shaker at 4 °C. The next day, membranes were washed four times with TBS-T for 10 min each, before adding secondary antibody for at least 2 h. Membranes were then washed three times for 5 min each with TBS-T, and a fourth time with TBS.

Immunoblots were visualized using chemiluminescence. SuperSignal West Femto Maximum Sensitivity Substrate was prepared according to the manufacturer's manual and distributed equally over a transparent film. The membrane was rolled over the substrate to allow equal distribution of substrate on the membrane.

## Structure prediction and analysis

Structures of the isolated EFR kinase domain or the intracellular domain were predicted using AlphaFold2 (*Jumper et al., 2021*) running it in Google CoLab (*Mirdita et al., 2022*). PDB files were downloaded and the best model (highest pLDDT score) was visualized in ChimeraX 1.6 (*Pettersen et al., 2021*). Hydrogen bonds were predicted in ChimeraX which uses angle and distance cutoffs for H-bonds described in *Mills and Dean, 1996*.

## Phylogenetic analysis and tree visualization

Multiple sequence alignments were retrieved from a previous phylogenetic study of plant LRR-RKs (*Dufayard et al., 2017*). Specifically, the trimmed multiple sequence alignment for subfamily XIIa was retrieved. The retrieved MSA was used for building a phylogenetic tree using the IQ-TREE webserver (*Kalyaanamoorthy et al., 2017*; *Minh et al., 2020*; *Trifinopoulos et al., 2016*). The generated tree file was then used to visualize the tree in R using the ggtree package v3.8.2 (*Yu et al., 2017*).

## Statistical analysis

In general, non-parametric Kruskal-Wallis tests with Dunn's post-hoc test (Benjamin-Hochberg correction) where applied in this study because either sample size were small, data was not normally distributed or variance between groups were not similar. Outliers shown in boxplots were not removed prior to statistical analysis.

## Acknowledgements

We thank Tamaryn Ellick for plant care, Ursin Stirnemann for help establishing the Rap-induced dimerization work, and Fabian Lachmann for help with preparative tasks. Further, we thank Jeonghayng Park, Eunkyoo Oh and Jungmook Kim for sharing materials of and information about the Rap-induced dimerization system. All past and current members of the Zipfel group are thanked for fruitful discussions. This project was funded by the University of Zürich (C.Z.), the Swiss National Science Foundation grant no. 31003 A_182625 (C.Z.), a joint European Research Area Network for Coordinating Action in Plant Sciences (ERA-CAPS) grant ('SICOPID') from UK Research and Innovation (BB/S004734/1) (C.Z.), and by NIH grant R35-GM122485 (M.A.L.).

## Additional information

### Funding

| Funder | Grant reference number | Author |
|---|---|---|
| Schweizerischer Nationalfonds zur Förderung der Wissenschaftlichen Forschung | 31003A_182625 | Cyril Zipfel |
| UK Research and Innovation | BB/S004734/1 | Cyril Zipfel |
| National Institute of General Medical Sciences | R35-GM122485 | Mark A Lemmon |

The funders had no role in study design, data collection and interpretation, or the decision to submit the work for publication.

### Author contributions

Henning Mühlenbeck, Investigation, Visualization, Methodology, Writing - original draft, Writing - review and editing; Yuko Tsutsui, Supervision, Investigation, Visualization, Methodology, Writing - review and editing; Mark A Lemmon, Supervision, Funding acquisition, Writing - review and editing; Kyle W Bender, Conceptualization, Supervision, Writing - review and editing; Cyril Zipfel, Conceptualization, Supervision, Project administration, Writing - review and editing

### Author ORCIDs

Kyle W Bender  http://orcid.org/0000-0002-1805-8097
Cyril Zipfel  http://orcid.org/0000-0003-4935-8583

Reviewer #1 (Public Review): https://doi.org/10.7554/eLife.92110.4.sa1
Reviewer #2 (Public Review): https://doi.org/10.7554/eLife.92110.4.sa2
Reviewer #3 (Public Review): https://doi.org/10.7554/eLife.92110.4.sa3
Author response https://doi.org/10.7554/eLife.92110.4.sa4

## Additional files

### Supplementary files

• MDAR checklist

### Data availability

All HDX RAW data for wild type and Y836F EFR have been deposited to the ProteomeXchange Consortium via ProteomeXchage via the PRIDE (*Perez-Riverol et al., 2022*) with the dataset identifier PXD049215. Supplemental information, including plasmid maps, replication information, cropping information, gene IDs, protein sequences, and PDB files, are available through a Zenodo repository (https://doi.org/10.5281/zenodo.10577812).

The following datasets were generated:

| Author(s) | Year | Dataset title | Dataset URL | Database and Identifier |
|---|---|---|---|---|
| Tsutsui Y | 2024 | Allosteric activation of the co-receptor BAK1 by the EFR receptor kinase initiates immune signaling | http://www.ebi.ac.uk/pride/archive/projects/PXD049215 | PRIDE, PXD049215 |
| Mühlenbeck H, Tsutsui Y, Lemmon MA, Bender KW, Zipfel C | 2024 | Supplemental Information - Allosteric activation of the co-receptor BAK1 by the EFR receptor kinase initiates immune signaling | https://doi.org/10.5281/zenodo.10577812 | Zenodo, 10.5281/zenodo.10577812 |

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

# Appendix 1.

## Appendix 1—key resources table

| Reagent type (species) or resource | Designation | Source or reference | Identifiers | Additional information |
|---|---|---|---|---|
| Strain, strain background (*Arabidopsis thaliana*) | Col-0 | ther | | wild-type ecotype |
| Strain, strain background (*Nicotiana benthamiana*) | WT | Lab stock | | Wild-type strain |
| strain, strain background (*Escherichia coli*) | BL21(DE3) pLPP | Amid Biosciences | Cat#: BLPP-201 | Co-expression of lambda phosphatase |
| Strain, strain background (*Escherichia coli*) | BL21(DE3)-V2R-pACYC LamP | PMID: 20436473 | | Co-expression of lambda phosphatase |
| Strain, strain background (*Agrobacterium tumefaciens*) | GV3101 | Lab stock | | |
| Strain, strain background (*Arabidopsis thaliana*) | *efr-1* | PMID: 16713565 | | |
| Genetic reagent (*Arabidopsis thaliana*) | EFR-GFP | This paper | | See Materials and methods |
| Genetic reagent (*Arabidopsis thaliana*) | F761H#2 | This paper | | See Materials and methods |
| Genetic reagent (*Arabidopsis thaliana*) | F761H#7 | This paper | | See Materials and methods |
| Genetic reagent (*Arabidopsis thaliana*) | Y836F#1 | This paper | | See Materials and methods |
| Genetic reagent (*Arabidopsis thaliana*) | Y836F#4 | This paper | | See Materials and methods |
| Genetic reagent (*Arabidopsis thaliana*) | F761H/Y836F#1 | This paper | | See Materials and methods |
| Genetic reagent (*Arabidopsis thaliana*) | F761H/Y836F#5 | This paper | | See Materials and methods |
| Genetic reagent (*Arabidopsis thaliana*) | SSAA#2 | This paper | | See Materials and methods |
| Genetic reagent (*Arabidopsis thaliana*) | SSAA#7 | This paper | | See Materials and methods |
| Genetic reagent (*Arabidopsis thaliana*) | F761H/SSAA#2 | This paper | | See Materials and methods |
| Genetic reagent (*Arabidopsis thaliana*) | F761H/SSAA#7 | This paper | | See Materials and methods |
| Sequence-based reagent | EFR-f | This paper | PCR primer | atcgGAAGACaaAATGA AGCTGTCCTTTTCACTTG |
| Sequence-based reagent | EFR Y836F-f | This paper | PCR primer | ggagtTtctgcacgtt cattgtcatgaccctgta |
| Sequence-based reagent | EFR SSAA-f | This paper | PCR primer | Gtttgctgctgctggt gtcagaggcaccat |
| Sequence-based reagent | EFR F761H-f | This paper | PCR primer | acgatgtcgtataccttgt gggtttcacattccgccata |
| Sequence-based reagent | EFR F761M-f | This paper | PCR primer | tacgatgtcgtataccttc atggtttcacattccgccata |
| Sequence-based reagent | EFR V845F-f | This paper | PCR primer | gcttaatatcacagtgagc gaaagggtcatgacaatgaacgtg |
| Sequence-based reagent | EFR L743F-f | This paper | PCR primer | gctttaggaggttgaaaac tttaaccgcgacgagttta |
| Sequence-based reagent | EFR L873E-f | This paper | PCR primer | ctcagGAGctctataaata cgatcgagaatcctttcta |
| Sequence-based reagent | EFR dNLLKH-f | This paper | PCR primer | taaagttttgggagcgacga aaagctttatggcggaatgtg |
| Sequence-based reagent | EFR D849N-f | This paper | PCR primer | cactgtAatattaagcca agcaacattcttctagacgat |

*Appendix 1 Continued on next page*

*Appendix 1 Continued*

| Reagent type (species) or resource | Designation | Source or reference | Identifiers | Additional information |
|---|---|---|---|---|
| Sequence-based reagent | BAK1-f | This paper | PCR primer | atcgGAAGACaaAATGGA ACGAAGATTAATGATCC |
| Sequence-based reagent | BAK1 Y403F-f | This paper | PCR primer | cttgcgtttttacatgat cattgcgacccaaaga |
| Sequence-based reagent | BAK1 I338H-f | This paper | PCR primer | gagatgCAtagtatggcg gttcacagaaacttgct |
| Sequence-based reagent | BAK1 D416N-f | This paper | PCR primer | catcgaAatgtgaaagctg caaatattttgttggatgaag |
| Sequence-based reagent | EFRprom-f | This paper | PCR primer | atcgGgtctctTACGCGTCTC aGTAGatctagacgat taagtaattgagca |
| Sequence-based reagent | BAK1prom-f | This paper | PCR primer | atcgGgtctctTACGCGTCTCa CCTAtgtcgtgaaaagggcac |
| Sequence-based reagent | XIIa2_Tmicd-f | This paper | PCR primer | atcgGCTCTTCgAAGGTT CTTCTACCGGTTCTGTTAT |
| Sequence-based reagent | XIIa3_Tmicd-f | This paper | PCR primer | atcgGCTCTTCgAAGG TTGCGATTGGGGTCA |
| Sequence-based reagent | XIIa4_Tmicd-f | This paper | PCR primer | atcgGCTCTTCgAAGATAA TCACCATTTGTGTCAGTG |
| Sequence-based reagent | XIIa5_Tmicd-f | This paper | PCR primer | atcgGCTCTTCgAAGGTT GTGATTGGAGTTAGCGT |
| Sequence-based reagent | XIIa6_Tmicd-f | This paper | PCR primer | atcgGCTCTTCgAAGGTTG CAATTTTAGTAAGCATAGG |
| Sequence-based reagent | FLS2_Tmicd-f | This paper | PCR primer | atcgGCTCTTCgAAGGTCA TCCTGATTATTCTTGGATCA GCCGCGGCaCTTCTTCTTGTCCTG |
| Sequence-based reagent | AlXIIa_D850N-f | This paper | PCR primer | atcgGCTCTTCAGTa ATATTAAGCCAAGCAACG |
| Sequence-based reagent | BrXIIa_D846N-f | This paper | PCR primer | atcgGCTCTTCAGTaA TCTTAAGCCAAGCAAC |
| Sequence-based reagent | GmXIIa_D829N-f | This paper | PCR primer | atcgGCTCTTCAGTaATA TTAAGCCAAGCAACATT |
| Sequence-based reagent | PtXIIa_D848N-f | This paper | PCR primer | atcgGCTCTTCAGT aatctgaagccaagcaa |
| Sequence-based reagent | SlXIIa_D883N-f | This paper | PCR primer | atcgGCTCTTCAGTaATAT AAAACCACAGAACATTCT |
| Sequence-based reagent | XIIa2_D803N-f | This paper | PCR primer | atcgGCTCTTCAGTaATC TCAAACCGAGCAATATCC |
| Sequence-based reagent | XIIa3_D838N-f | This paper | PCR primer | atcgGCTCTTCACaATCT TAAGCCAAGCAACATACT |
| Sequence-based reagent | XIIa4_D856N-f | This paper | PCR primer<br><br>atcgGCTCTTCAGTaATATT AAGCCAAGCAATATTCTACTA | |
| Sequence-based reagent | XIIa5_D839N-f | This paper | PCR primer | atcgGCTCTTCAGCaA TCTTAAGCCAAGCAACGT |
| Sequence-based reagent | XIIa6_D840N-f | This paper | PCR primer | atcgGCTCTTCAGCaATC TCAAGCCAAGCAACG |
| Sequence-based reagent | FLS2icd_D997N-f | This paper | PCR primer | atcgGCTCTTCAGTaATC TGAAGCCAGCTAATATACT |
| Sequence-based reagent | FLS2_D997N-f | This paper | PCR primer | GTTCATTGTaATCTGAAGC CAGCTAATATACTCCTTGACA |
| Sequence-based reagent | EFR_ecto-f | This paper | PCR primer | atcgGGTCTCaAATGAAG CTGTCCTTTTCACTTG |
| Sequence-based reagent | EFR_SapIdom-f | This paper | PCR primer | GTTATGAgGAGCTTCAT AGTGCAACAAGTCGCTTC |
| Sequence-based reagent | EFR_Esp3Idom-f | This paper | PCR primer | TTCCCGTgTCTTTCGGG AAGCTTTTGAACTTGC |

*Appendix 1 Continued on next page*

*Appendix 1 Continued*

| Reagent type (species) or resource | Designation | Source or reference | Identifiers | Additional information |
|---|---|---|---|---|
| Sequence-based reagent | ccdB cassette-f | This paper | PCR primer | atgcGGTCTCAGAAGgGAAG AGCAAAGCTGAACGA GAAACGTAAAAT |
| Sequence-based reagent | pET28_lin-f | This paper | PCR primer | tgagatccggctgctaa |
| Sequence-based reagent | pETGG cloning cassette-f | This paper | PCR primer | AGAAGGAGATATACCAA TGAGAGACCAAAGCTGAA |
| Sequence-based reagent | P641-BsaI_mut1-f | This paper | PCR primer | agagacCaaagctgaacga gaaacgtaaaatgatataaata |
| Sequence-based reagent | P641-BsaI_mut2-f | This paper | PCR primer | gccagtGgtctcttc tggtcgtgactggg |
| Sequence-based reagent | 6xHis-FKBP-f | This paper | PCR primer | atcgCGTCTCtTACGGGTCT CaAATGggcagcagccatcatc atcatcacagcagcggcGGA GTGCAGGTGGAAAC |
| Sequence-based reagent | 6xHis-FRB-f | This paper | PCR primer | atcgCGTCTCtTACGGGTCTCaA ATGggcagcagccatcatcatcatc acagcagcggcATCCT CTGGCATGAGATGT |
| Sequence-based reagent | EFR_icd (stop)-f | This paper | PCR primer | atcgCGTCTCtTACGGGTCT CaAGGTaagaggaaaaag aaaaacaatgcc |
| Sequence-based reagent | EFR_icd (no stop)-f | This paper | PCR primer | atcgCGTCTCtTACGGGTCTCa AGGTaagaggaaa aagaaaaacaatgcc |
| Sequence-based reagent | BAK1_icd (stop)-f | This paper | PCR primer | atcgCGTCTCtTACGGGTCTC aAGGTgcttggtggcgaagg |
| Sequence-based reagent | FLS2_icd (no stop)-f | This paper | PCR primer | atcgCGTCTCtTACGGGTCTCaAG GTACCTGTTGCAAGAAA AAAGAAAAAAAG |
| Sequence-based reagent | BRI1_icd (no stop)-f | This paper | PCR primer | atcgGGTCTCaAGGT GGTAGAGAGATGA GGAAGAGA |
| Sequence-based reagent | BRI1icd_D1009N-f | This paper | PCR primer | atcgGGTCTCaAaACATG AAATCCAGTAATGTGTTGC |
| Sequence-based reagent | EFR-r | This paper | PCR primer | cgatGAAGACttCGAAcc CATAGTATGCATGTCCG |
| Sequence-based reagent | EFR Y836F-r | This paper | PCR primer | gtgcagaAactccaaagc tgaagccacatctat |
| Sequence-based reagent | EFR SSAA-r | This paper | PCR primer | gcagcagcaaactggttt agaaaggattctcgatcg |
| Sequence-based reagent | EFR F761H-r | This paper | PCR primer | tatggcggaatgtgaaacc cacaagggtatacgacatcgt |
| Sequence-based reagent | EFR F761M-r | This paper | PCR primer | tatggcggaatgtgaaaccat gaagggtatacgacatcgta |
| Sequence-based reagent | EFR V845F-r | This paper | PCR primer | cacgttcattgtcatgaccctt tcgctcactgtgatattaagc |
| Sequence-based reagent | EFR L743F-r | This paper | PCR primer | taaactcgtcgcggttaaa gttttcaacctcctaaagc |
| Sequence-based reagent | EFR L873E-r | This paper | PCR primer | tatagagCTCctgagcca aaccaaagtcactaacatg |
| Sequence-based reagent | EFR dNLLKH-r | This paper | PCR primer | tcgtcgctcccaaaactttaa ccgcgacgagtttattct |
| Sequence-based reagent | EFR D849N-r | This paper | PCR primer | ggcttaatatTacagtgag ctacagggtcatgaca |
| Sequence-based reagent | BAK1-r | This paper | PCR primer | cgatGAAGACttAAGCc cTTATCTTGGACCCGAGG |
| Sequence-based reagent | BAK1 Y403F-r | This paper | PCR primer | catgtaaaaacgca agccctcttgcagatccca |

*Appendix 1 Continued on next page*

*Appendix 1 Continued*

| Reagent type (species) or resource | Designation | Source or reference | Identifiers | Additional information |
|---|---|---|---|---|
| Sequence-based reagent | BAK1 I338H-r | This paper | PCR primer | ccatactaTGcatctca acctctgtctggaactgc |
| Sequence-based reagent | BAK1 D416N-r | This paper | PCR primer | ctttcacatTtcgatgaata atctttgggtcgcaatg |
| Sequence-based reagent | EFRprom-r | This paper | PCR primer | atcgGgtctctCAGACGTCTC aCATTgtcgattataaaaa gataaaagaaaggtt |
| Sequence-based reagent | BAK1prom-r | This paper | PCR primer | atcgGgtctctCAGACGTCT CaCATTtttatcctcaagag attaaaaacaaac |
| Sequence-based reagent | XIIa2_Tmicd-r | This paper | PCR primer | cgatGCTCTTCtCGAAccT GAACTAGCTTCTCCTTGTG |
| Sequence-based reagent | XIIa3_Tmicd-r | This paper | PCR primer | cgatGCTCTTCtCGAAccA CGTCTGGCTGTTCTCC |
| Sequence-based reagent | XIIa4_Tmicd-r | This paper | PCR primer | cgatGCTCTTCtCGAAccAG TCTCCTCGTCTCTGAAA |
| Sequence-based reagent | XIIa5_Tmicd-r | This paper | PCR primer | cgatGCTCTTCtCGAAccACGCC AAGTCGTTCTACTGGCTT TAAAGAACCTCTCTCTGAT TGAGATCAACTCCTTG |
| Sequence-based reagent | XIIa6_Tmicd-r | This paper | PCR primer | cgatGCTCTTCtCGAAccAC GTCTAGGTGTTCTTCTG |
| Sequence-based reagent | FLS2_Tmicd-r | This paper | PCR primer | cgatGCTCTTCtCGAAccA ACTTCTCGATCCTCGTTAC |
| Sequence-based reagent | AlXIIa_D850N-r | This paper | PCR primer | atcgGCTCTTCAtACA GTGAGCTACAGGGT |
| Sequence-based reagent | BrXIIa_D846N-r | This paper | PCR primer | atcgGCTCTTCAtACAG TGAGCTATTTGGTCAT |
| Sequence-based reagent | GmXIIa_D829N-r | This paper | PCR primer | atcgGCTCTTCAtACA GTGAACTACGGCCT |
| Sequence-based reagent | PtXIIa_D848N-r | This paper | PCR primer | atcgGCTCTTCAtACa atgaatgatgggcatg |
| Sequence-based reagent | SlXIIa_D883N-r | This paper | PCR primer | atcgGCTCTTCAtACA GTGAATCATGGGTGTT |
| Sequence-based reagent | XIIa2_D803N-r | This paper | PCR primer | atcgGCTCTTCAtACAGT GAACAACTTTTACAGGTGA |
| Sequence-based reagent | XIIa3_D838N-r | This paper | PCR primer | atcgGCTCTTCATtGCAA TGAGCTATAGGCTCATGA |
| Sequence-based reagent | XIIa4_D856N-r | This paper | PCR primer | atcgGCTCTTCAtACA GTGGGCTATAGGGTTGT |
| Sequence-based reagent | XIIa5_D839N-r | This paper | PCR primer | atcgGCTCTTCAtGCAA TGAGCTATAGGTTCATGACA |
| Sequence-based reagent | XIIa6_D840N-r | This paper | PCR primer | atcgGCTCTTCAtGCAAT GAGCTATAGGCTCATGAC |
| Sequence-based reagent | FLS2icd_D997N-r | This paper | PCR primer | atcgGCTCTTCAtACAA TGAACGATGGGAAAACC |
| Sequence-based reagent | FLS2_D997N-r | This paper | PCR primer | CTTCAGATtACAATGAAC GATGGGAAAACCATATCCAGA |
| Sequence-based reagent | EFR_ecto-r | This paper | PCR primer | atcgGGTCTCacTTCT TTCTAACTGACAGAGGC |
| Sequence-based reagent | EFR_SapIdom-r | This paper | PCR primer | TGAAGCTCcTCATAACT TACCTTCTCATGGAACATCC |
| Sequence-based reagent | EFR_Esp3Idom-r | This paper | PCR primer | GAAAGAcACGGGAAGT TCTCCACTCAACATATTTGTTT |
| Sequence-based reagent | ccdB cassette-r | This paper | PCR primer | tacgGGTCTCACGAAGA AGAGCactggctgtgtataaggga |
| Sequence-based reagent | pET28_lin-r | This paper | PCR primer | ggtatatctccttct taaagttaaaca |

*Appendix 1 Continued on next page*

*Appendix 1 Continued*

| Reagent type (species) or resource | Designation | Source or reference | Identifiers | Additional information |
|---|---|---|---|---|
| Sequence-based reagent | pETGG cloning cassette-r | This paper | PCR primer | AGCAGCCGGATCTC AAAGCAGAGACCACTGGC |
| Sequence-based reagent | P641-Bsal_mut1-r | This paper | PCR primer | cagctttGgtctctcg tacggcctcctgt |
| Sequence-based reagent | P641-Bsal_mut2-r | This paper | PCR primer | agagacCactggctgtg tataagggagcctga |
| Sequence-based reagent | 6xHis-FKBP-r | This paper | PCR primer | atcgCGTCTCtCAGAGGTC TCaACCTGAGCCGCTTTCC |
| Sequence-based reagent | 6xHis-FRB-r | This paper | PCR primer | atcgCGTCTCtCAGAGGTCT CaACCTGAGCCGCTCTTT |
| Sequence-based reagent | EFR_icd (stop)-r | This paper | PCR primer | atcgCGTCTCtCAGAGGTCTC aAAGCttacatagtatgcatgtccgtatt |
| Sequence-based reagent | EFR_icd (no stop)-r | This paper | PCR primer | atcgCGTCTCtCAGAGGTCTC aAAGCttacatagtatgcatgtccgtatt |
| Sequence-based reagent | BAK1_icd (stop)-r | This paper | PCR primer | atcgCGTCTCtCAGAGGTC TCaAAGCttatcttggacccgaggg |
| Sequence-based reagent | FLS2_icd (no stop)-r | This paper | PCR primer | atcgCGTCTCtCAGAGGTCTCaC GAAccAACTTCTCGATCCTCGTTACG |
| Sequence-based reagent | BRI1_icd (no stop)-r | This paper | PCR primer | atcgGGTCTCaCGAAccTAA TTTTCCTTCAGGAACTTCTTTTATA |
| Sequence-based reagent | BRI1icd_D1009N-r | This paper | PCR primer | atcgGGTCTCaGTtTCTGT GGATGATATGCGGAC |
| Antibody | GFP-HRP; mouse monoclonal | Santa Cruz Biotechnology | Cat#: sc-9996 | Dilution: 1:1000 |
| Antibody | BAK1; rabbit polyclonal | PMID: 21693696 | | Dilution: 1:10000 |
| Antibody | BAK1 pS612; rabbit polyclonal | PMID: 30177827 | | Dilution: 1:2000 |
| Antibody | FKBP12; mouse monoclonal | Santa Cruz Biotechnology | Cat#: sc-133067 | Dilution: 1:500 |
| Antibody | p44/42; rabbit polyclonal | Cell Signaling Technology | Cat#: 9101 | Dilution: 1:1000 |
| Antibody | Anti-rabbit-HRP; goat polyclonal | Sigma-Aldrich | Cat#: A0545 | Dilution: 1:10000 |
| Antibody | Anti-mouse-HRP; goat polyclonal | Sigma-Aldrich | Cat#: A0168 | Dilution: 1:5000 |
| Recombinant DNA reagent | p641-EFRprom(Bsal) | This paper | | See Materials and methods |
| Recombinant DNA reagent | 35 S_omegaEnhancer | PMID: 21364738 | | |
| Recombinant DNA reagent | pICH41258-nMyr-FKBP | PMID: 33964457 | | |
| Recombinant DNA reagent | pICH41258-nMyr-FRB | PMID: 33964457 | | |
| Recombinant DNA reagent | pICSL01003-GFP | PMID: 21364738 | | |
| Recombinant DNA reagent | pICSL01003-mEGFP | Mark Youles, TSL | | |
| Recombinant DNA reagent | HSP18t | Mark Youles, TSL | | |
| Recombinant DNA reagent | p641-6xHis-FKBP | This paper | | See Materials and methods |
| Recombinant DNA reagent | p641-6xHis-FRB | This paper | | See Materials and methods |
| Recombinant DNA reagent | pICSL01005-EFR | This paper | | See Materials and methods |
| Recombinant DNA reagent | pICSL01005-EFR Y836F | This paper | | See Materials and methods |
| Recombinant DNA reagent | pICSL01005-EFR SSAA | This paper | | See Materials and methods |
| Recombinant DNA reagent | pICSL01005-EFR F761H | This paper | | See Materials and methods |
| Recombinant DNA reagent | pICSL01005-EFR F761M | This paper | | See Materials and methods |
| Recombinant DNA reagent | pICSL01005-EFR dNLLKH | This paper | | See Materials and methods |
| Recombinant DNA reagent | pICSL01005-EFR L743F | This paper | | See Materials and methods |
| Recombinant DNA reagent | pICSL01005-EFR L873E | This paper | | See Materials and methods |
| Recombinant DNA reagent | pICSL01005-EFR F761H Y836F | This paper | | See Materials and methods |
| Recombinant DNA reagent | pICSL01005-EFR F761M Y836F | This paper | | See Materials and methods |

*Appendix 1 Continued on next page*

*Appendix 1 Continued*

| Reagent type (species) or resource | Designation | Source or reference | Identifiers | Additional information |
|---|---|---|---|---|
| Recombinant DNA reagent | pICSL01005-EFR dNLLKH Y836F | This paper | | See Materials and methods |
| Recombinant DNA reagent | pICSL01005-EFR L743F Y836F | This paper | | See Materials and methods |
| Recombinant DNA reagent | pICSL01005-EFR L873E Y836F | This paper | | See Materials and methods |
| Recombinant DNA reagent | pICSL01005-EFR D849N | This paper | | See Materials and methods |
| Recombinant DNA reagent | pICSL01005-EFR F761H D849N | This paper | | See Materials and methods |
| Recombinant DNA reagent | pICSL01005-EFR F761H SSAA | This paper | | See Materials and methods |
| Recombinant DNA reagent | pICSL01005-EFR F761M SSAA | This paper | | See Materials and methods |
| Recombinant DNA reagent | pICH41308-BAK1 | This paper | | See Materials and MethodsSee Materials and methods |
| Recombinant DNA reagent | pICH41308-BAK1 Y403F | This paper | | See Materials and MethodsSee Materials and methods |
| Recombinant DNA reagent | pICSL86955 | PMID: 21364738 | | |
| Recombinant DNA reagent | pICSL86955-pEFR::EFR-GFP::HSP18t | This paper | | See Materials and methods |
| Recombinant DNA reagent | pICSL86955-pEFR::EFR Y836F-GFP::HSP18t | This paper | | See Materials and methods |
| Recombinant DNA reagent | pICSL86955-pEFR::EFR SSAA-GFP::HSP18t | This paper | | See Materials and methods |
| Recombinant DNA reagent | pICSL86955-pEFR::EFR F761H-GFP::HSP18t | This paper | | See Materials and methods |
| Recombinant DNA reagent | pICSL86955-pEFR::EFR F761M-GFP::HSP18t | This paper | | See Materials and methods |
| Recombinant DNA reagent | pICSL86955-pEFR::EFR dNLLKH-GFP::HSP18t | This paper | | See Materials and methods |
| Recombinant DNA reagent | pICSL86955-pEFR::EFR L743F-GFP::HSP18t | This paper | | See Materials and methods |
| Recombinant DNA reagent | pICSL86955-pEFR::EFR L873E-GFP::HSP18t | This paper | | See Materials and methods |
| Recombinant DNA reagent | pICSL86955-pEFR::EFR F761H Y836F-GFP::HSP18t | This paper | | See Materials and methods |
| Recombinant DNA reagent | pICSL86955-pEFR::EFR F761M Y836F-GFP::HSP18t | This paper | | See Materials and methods |
| Recombinant DNA reagent | pICSL86955-pEFR::EFR dNLLKH Y836F-GFP::HSP18t | This paper | | See Materials and methods |
| Recombinant DNA reagent | pICSL86955-pEFR::EFR L743F Y836F-GFP::HSP18t | This paper | | See Materials and methods |
| Recombinant DNA reagent | pICSL86955-pEFR::EFR Y836F L873E-GFP::HSP18t | This paper | | See Materials and methods |
| Recombinant DNA reagent | pICSL86955-pEFR::EFR F761H SSAA-GFP::HSP18t | This paper | | See Materials and methods |
| Recombinant DNA reagent | pICSL86955-pEFR::EFR F761M SSAA-GFP::HSP18t | This paper | | See Materials and methods |
| Recombinant DNA reagent | p641 | PMID: 31300661 | | |
| Recombinant DNA reagent | p641-EFR_icd | This paper | | See Materials and methods |
| Recombinant DNA reagent | p641-EFR F761H_icd | This paper | | See Materials and methods |
| Recombinant DNA reagent | p641-EFR D849N_icd | This paper | | See Materials and methods |
| Recombinant DNA reagent | p641-EFR F761H D849N_icd | This paper | | See Materials and methods |
| Recombinant DNA reagent | p641-BAK1_icd | This paper | | See Materials and methods |
| Recombinant DNA reagent | p641-BAK1 Y403F_icd | This paper | | See Materials and methods |
| Recombinant DNA reagent | p641-BAK1 D416N_icd | This paper | | See Materials and methods |
| Recombinant DNA reagent | pICSL86955-35S::EFRecto-ccdB-GFP::HSP18t | This paper | | See Materials and methods |

*Appendix 1 Continued on next page*

*Appendix 1 Continued*

| Reagent type (species) or resource | Designation | Source or reference | Identifiers | Additional information |
|---|---|---|---|---|
| Recombinant DNA reagent | pICSL86955-35S::EFRecto-XIIa2icd-mEGFP::HSP18t | This paper | | See Materials and methods |
| Recombinant DNA reagent | pICSL86955-35S::EFRecto-XIIa2 D803N-mEGFP::HSP18t | This paper | | See Materials and methods |
| Recombinant DNA reagent | pICSL86955-35S::EFRecto-XIIa3icd-mEGFP::HSP18t | This paper | | See Materials and methods |
| Recombinant DNA reagent | pICSL86955-35S::EFRecto-XIIa3 D838N-mEGFP::HSP18t | This paper | | See Materials and methods |
| Recombinant DNA reagent | pICSL86955-35S::EFRecto-XIIa4icd-mEGFP::HSP18t | This paper | | See Materials and methods |
| Recombinant DNA reagent | pICSL86955-35S::EFRecto-XIIa4 D856N-mEGFP::HSP18t | This paper | | See Materials and methods |
| Recombinant DNA reagent | pICSL86955-35S::EFRecto-XIIa5icd-mEGFP::HSP18t | This paper | | See Materials and methods |
| Recombinant DNA reagent | pICSL86955-35S::EFRecto-XIIa5 D839N-mEGFP::HSP18t | This paper | | See Materials and methods |
| Recombinant DNA reagent | pICSL86955-35S::EFRecto-XIIa6icd-mEGFP::HSP18t | This paper | | See Materials and methods |
| Recombinant DNA reagent | pICSL86955-35S::EFRecto-XIIa6 D840N-mEGFP::HSP18t | This paper | | See Materials and methods |
| Recombinant DNA reagent | pICSL86955-35S::EFRecto-FLS2icd-mEGFP::HSP18t | This paper | | See Materials and methods |
| Recombinant DNA reagent | pICSL86955-35S::EFRecto-FLS2 D997N-mEGFP::HSP18t | This paper | | See Materials and methods |
| Recombinant DNA reagent | pICSL86955-35S::EFRecto-EFRicd-mEGFP::HSP18t | This paper | | See Materials and methods |
| Recombinant DNA reagent | pICSL86955-35S::EFRecto-EFR D849N-mEGFP::HSP18t | This paper | | See Materials and methods |
| Recombinant DNA reagent | pICSL86955-35S::EFRecto-XA21icd-mEGFP::HSP18t | This paper | | See Materials and methods |
| Recombinant DNA reagent | pICSL86955-35S::EFRecto-XA21 D803N-mEGFP::HSP18t | This paper | | See Materials and methods |
| Recombinant DNA reagent | pICSL86955-35S::EFRecto-Aralyicd-mEGFP::HSP18t | This paper | | See Materials and methods |
| Recombinant DNA reagent | pICSL86955-35S::EFRecto-Araly D850N-mEGFP::HSP18t | This paper | | See Materials and methods |
| Recombinant DNA reagent | pICSL86955-35S::EFRecto-Brapaicd-mEGFP::HSP18t | This paper | | See Materials and methods |
| Recombinant DNA reagent | pICSL86955-35S::EFRecto-Brapa D846N-mEGFP::HSP18t | This paper | | See Materials and methods |
| Recombinant DNA reagent | pICSL86955-35S::EFRecto-Solycicd-mEGFP::HSP18t | This paper | | See Materials and methods |
| Recombinant DNA reagent | pICSL86955-35S::EFRecto-Solyc D883N-mEGFP::HSP18t | This paper | | See Materials and methods |
| Recombinant DNA reagent | pICSL86955-35S::EFRecto-Poptricd-mEGFP::HSP18t | This paper | | See Materials and methods |
| Recombinant DNA reagent | pICSL86955-35S::EFRecto-Poptr D848N-mEGFP::HSP18t | This paper | | See Materials and methods |
| Recombinant DNA reagent | pICSL86955-35S::EFRecto-Glymaicd-mEGFP::HSP18t | This paper | | See Materials and methods |
| Recombinant DNA reagent | pICSL86955-35S::EFRecto-GlymaD829N-mEGFP::HSP18t | This paper | | See Materials and methods |
| Recombinant DNA reagent | pETGG | This paper | | See Materials and methods |
| Recombinant DNA reagent | pETGG-6xHis-FKBP-EFRicd-mEGFP | This paper | | See Materials and methods |
| Recombinant DNA reagent | pETGG-6xHis-FKBP-EFRicd D849N-mEGFP | This paper | | See Materials and methods |

*Appendix 1 Continued on next page*

*Appendix 1 Continued*

| Reagent type (species) or resource | Designation | Source or reference | Identifiers | Additional information |
|---|---|---|---|---|
| Recombinant DNA reagent | pETGG-6xHis-FKBP-BRI1-mEGFP | This paper | | See Materials and methods |
| Recombinant DNA reagent | pETGG-6xHis-FKBP-BRI1 D1009N-mEGFP | This paper | | See Materials and methods |
| Recombinant DNA reagent | pETGG-6xHis-FKBP-FLS2-mEGFP | This paper | | See Materials and methods |
| Recombinant DNA reagent | pETGG-6xHis-FKBP-FLS D997N-mEGFP | This paper | | See Materials and methods |
| Recombinant DNA reagent | pETGG-6xHis-FKBP-XIIa5-mEGFP | This paper | | See Materials and methods |
| Recombinant DNA reagent | pETGG-6xHis-FKBP-XIIa5 D839N-mEGFP | This paper | | See Materials and methods |
| Recombinant DNA reagent | pETGG-6xHis-FRB-BAK1 | This paper | | See Materials and methods |
| Recombinant DNA reagent | pET28a(+)–6xHis-BIK1 D202N | This paper | | See Materials and methods |
| Recombinant DNA reagent | pC1a1-35S::FRB-BAK1_icd::HSP18t | This paper | | See Materials and methods |
| Recombinant DNA reagent | pC1a1-35S::FRB-BAK1 Y403F_icd::HSP18t | This paper | | See Materials and methods |
| Recombinant DNA reagent | pC1a1-35S::FRB-BAK1 D416N_icd::HSP18t | This paper | | See Materials and methods |
| Recombinant DNA reagent | pC1a2-35S::FKBP-EFR_icd::HSP18t | This paper | | See Materials and methods |
| Recombinant DNA reagent | pC1a2-35S::FKBP-EFR F761H_icd::HSP18t | This paper | | See Materials and methods |
| Recombinant DNA reagent | pICSL4723 | PMID: 21364738 | | See Materials and methods |
| Recombinant DNA reagent | pICSL4723-FRB-BAK1-FKBP-EFR | This paper | | See Materials and methods |
| Recombinant DNA reagent | pICSL4723-FRB-BAK1 D416N-FKBP-EFR | This paper | | See Materials and methods |
| Recombinant DNA reagent | pICSL4723-FRB-BAK1-FKBP-EFR F761H | This paper | | See Materials and methods |
| Recombinant DNA reagent | pICSL4723-FRB-BAK1 D416N-FKBP-EFR F761H | This paper | | See Materials and methods |
| Recombinant DNA reagent | pICSL4723-FRB-BAK1 Y403F-FKBP-EFR | This paper | | See Materials and methods |
| Recombinant DNA reagent | pICSL4723-FRB-BAK1 Y403F-FKBP-EFR F761H | This paper | | See Materials and methods |

