## [Editor Report · eLife assessment]

This manuscript reports **important** in vitro biochemical and in planta experiments to study the receptor activation mechanism of plant membrane receptor kinase complexes through the non-catalytic function of an active protein kinase. Several lines of evidence **convincingly** show that one such receptor kinase with pseudokinase-like function, the immune receptor EFR achieves an active conformation following phosphorylation by a co-receptor kinase, and then in turn activates the co-receptor kinase allosterically to enable it to phosphorylate down-stream signaling components. This manuscript will be of interest to scientists focusing on cell signalling and allosteric regulation.

---

## [Referee Report · Reviewer #1 (Public Review)]

Summary

The authors use an elegant but somewhat artificial heterodimerisation approach to activate the isolated cytoplasmic domains of different receptor kinases (RKs) including the receptor kinase BRI1 and EFR. The developmental RK BRI1 is known to be activated by the co-receptor BAK1. Active BRI1 is then able to phosphorylate downstream substrates. The immune receptor EFR is also an active protein kinase also activated by the co-receptor BAK1. EFR however appears to have little or no kinase activity but seems to use an allosteric mechanism to in turn enable BAK1 to phosphorylate the substrate kinase BIK1. EFR tyrosine phosphorylation by BAK1 appears to trigger a conformational change in EFR, activating the receptor. Likewise, kinase activating mutations can cause similar conformational transitions in EFR and also in BAK1 in vitro and in planta.

Strengths:

I particularly liked The HDX experiments coupled with mutational analysis (Fig. 2) and the design and testing of the kinase activating mutations (Fig. 3), as they provide novel mechanistic insights into the activation mechanisms of EFR and of BAK1. These findings are nicely extended by the large-scale identification of EFR-related RKs from different species with potentially similar activation mechanisms (Fig. 5).

Overall this is an interesting study that aims to advance our understanding of the activation mechanisms of different plant receptor kinases with important functions in plant immunity.

---

## [Referee Report · Reviewer #2 (Public Review)]

Summary:

Transmembrane signaling in plants is crucial for homeostasis. In this study, the authors set out to understand to what extent catalytic activity in the EFR tyrosine kinase is required in order to transmit a signal. This work was driven by mounting data that suggest many eukaryotic kinases do not rely on catalysis for signal transduction, relying instead on conformational switching to relay information. The crucial findings reported here involve the realisation that a kinase-inactive EFR can still activate (ie lead to downstream phosphorylation) of its partner protein BAK1. Using a convincing set of biochemical, mass spectrometric (HD-exchange) and in vivo assays, the team suggests a model in which EFR is likely phosphorylated in the canonical activation segment (where two Ser residues are present), which is sufficient to generate a conformation that can activate BAK1 through dimersation. A model is put forward involving C-helix positioning in BAK1, and the model is extended to other 'non-RD' kinases in Arabidopsis kinases that likely do not require kinase activity for signaling.

Strengths:

The work uses logical and well-controlled approaches throughout, and is clear and convincing in most areas, linking data from IPs, kinase assays (including clear 32P-based biochemistry), HD-MX data (from non-phosphorylated EFR) structural biology, oxidative burst data and infectivity assays. Repetitions and statistical analysis all appear appropriate.

Overall, the work builds a convincing story and the discussion does a clear job of explaining the potential impact of these findings (and perhaps an explanation of why so many Arabidopsis kinases are 'pseudokinases', including XPS1 and XIIa6, where this is shown explicitly).

Impact:

The work is an important new step in the huge amount of follow-up work needed to examine how kinases and pseudokinases 'talk' to each other in (especially) the plant kingdom, where significant genetic expansions have occurred. The broader impact is that we might understand better how to manipulate signaling for the benefit of plants and mankind; as the authors suggest, their study is a natural progression both of their own work and the kingdom-wide study of the Kannan group.

---

## [Referee Report · Reviewer #3 (Public Review)]

The study presents strong evidence for allosteric activation of plant receptor kinases, which enhances our understanding of the non-catalytic mechanisms employed by this large family of receptors.

Plant receptor kinases (RKs) play a critical role in transducing extracellular signals. The activation of RKs involves homo- or heterodimerization of the RKs, and it is believed that mutual phosphorylation of their intracellular kinase domains initiates downstream signaling. However, this model faces a challenge in cases where the kinase domain exhibits pseudokinase characteristics. In their recent study, Mühlenbeck et al. reveal the non-catalytic activation mechanisms of the EFR-BAK1 complex in plant receptor kinase signaling. Specifically, they aimed to determine that the EFR kinase domain activates BAK1 not through its kinase activity, but rather by utilizing a "conformational toggle" mechanism to enter an active-like state, enabling allosteric trans-activation of BAK1. The study sought to elucidate the structural elements and mutations of EFR that affect this conformational switch, as well as explore the implications for immune signaling in plants. To investigate the activation mechanisms of the EFR-BAK1 complex, the research team employed a combination of mutational analysis, structural studies, and hydrogen-deuterium exchange mass spectrometry (HDX-MS) analysis. For instance, through HDX-MS analysis, Mühlenbeck et al. discovered that the EFR (Y836F) mutation impairs the accessibility of the active-like conformation. On the other hand, they identified the EFR (F761H) mutation as a potent intragenic suppressor capable of stabilizing the active-like conformation, highlighting the pivotal role of allosteric regulation in BAK1 kinase activation. The data obtained from this methodology strengthens their major conclusion. Moreover, the researchers propose that the allosteric activation mechanism may extend beyond the EFR-BAK1 complex, as it may also be partially conserved in the Arabidopsis LRR-RK XIIa kinases. This suggests a broader role for non-catalytic mechanisms in plant RK signaling.

The allosteric activation mechanism was demonstrated for receptor tyrosine kinases (RTKs) many years ago. A similar mechanism has been suggested for the activation of plant RKs, but experimental evidence for this conclusion is lacking. Data in this study represent a significant advancement in our understanding of non-catalytic mechanisms in plant RK signaling. By shedding light on the allosteric regulation of BAK1, the study provides a new paradigm for future research in this area.

---

## [Author Response]

The following is the authors’ response to the previous reviews.

**eLife assessment**
This manuscript reports important in vitro biochemical and in planta experiments to study the receptor activation mechanism of plant membrane receptor kinase complexes with non-catalytic intracellular kinase domains. Several lines of evidence convincingly show that one such putative pseudokinase, the immune receptor EFR achieves an active conformation following phosphorylation by a co-receptor kinase, and then in turn activates the co-receptor kinase allosterically to enable it to phosphorylate down-stream signaling components. This manuscript will be of interest to scientists focusing on cell signalling and allosteric regulation.

We wish to clarify that EFR is itself, not a pseudokinase. We could show in previous work (Bender et al., 2021; https://doi.org/10.1073/pnas.2108242118 ) that EFR has catalytic activity in vitro. This catalytic activity is, however, not required for elf18-induced immune signaling in planta.

**Public Reviews:**

**Reviewer #1 (Public Review):**
SummaryThe authors use an elegant but somewhat artificial heterodimerisation approach to activate the isolated cytoplasmic domains of different receptor kinases (RKs) including the receptor kinase BRI1 and EFR. The developmental RK BRI1 is known to be activated by the co-receptor BAK1. Active BRI1 is then able to phosphorylate downstream substrates. The immune receptor EFR is also an active protein kinase also activated by the co-receptor BAK1. EFR however appears to have little or no kinase activity but seems to use an allosteric mechanism to in turn enable BAK1 to phosphorylate the substrate kinase BIK1. EFR tyrosine phosphorylation by BAK1 appears to trigger a conformational change in EFR, activating the receptor. Likewise, kinase activating mutations can cause similar conformational transitions in EFR and also in BAK1 in vitro and in planta.

We wish to clarify that we make no strong link between tyrosine phosphorylation and the conformational change leading to activation of the complex. Rather, the HDX-MS data demonstrate the structural importance of Tyr836 for the activation mechanism. At present, we do not know how phosphorylation of the residue would affect the activation process.

Strengths:I particularly liked The HDX experiments coupled with mutational analysis (Fig. 2) and the design and testing of the kinase activating mutations (Fig. 3), as they provide novel mechanistic insights into the activation mechanisms of EFR and of BAK1. These findings are nicely extended by the large-scale identification of EFR-related RKs from different species with potentially similar activation mechanisms (Fig. 5).Weaknesses:In my opinion, there are currently two major issues with the present manuscript. (1) The authors have previously reported that the EFR kinase activity is dispensible for immune signaling (https://pubmed.ncbi.nlm.nih.gov/34531323/) but the wild-type EFR receptor still leads to a much better phosphorylation of the BIK1 substrate when compared to the kinase inactive D849N mutant protein (Fig. 1). (2) How the active-like conformation of EFR is in turn activating BAK1 is poorly characterized, but appears to be the main step in the activation of the receptor complex. Extending the HDX analyses to resting and Rap-activated receptor complexes could be a first step to address this question, but these HDX studies were not carried out due to technical limitations.Overall this is an interesting study that aims to advance our understanding of the activation mechanisms of different plant receptor kinases with important functions in plant immunity.
**Reviewer #2 (Public Review):**
Summary:Transmembrane signaling in plants is crucial for homeostasis. In this study, the authors set out to understand to what extent catalytic activity in the EFR tyrosine kinase is required in order to transmit a signal. This work was driven by mounting data that suggest many eukaryotic kinases do not rely on catalysis for signal transduction, relying instead on conformational switching to relay information. The crucial findings reported here involve the realisation that a kinase-inactive EFR can still activate (ie lead to downstream phosphorylation) of its partner protein BAK1. Using a convincing set of biochemical, mass spectrometric (HD-exchange) and in vivo assays, the team suggest a model in which EFR is likely phosphorylated in the canonical activation segment (where two Ser residues are present), which is sufficient to generate a conformation that can activate BAK1 through dimersation. A model is put forward involving C-helix positioning in BAK1, and the model extended to other 'non-RD' kinases in Arabidopsis kinases that likely do not require kinase activity for signaling.

We prefer not to describe EFR as a tyrosine kinase. It may be the case that EFR can function under certain conditions as a dual-specificity protein kinase, but this has never been demonstrated experimentally. We therefore describe EFR as a Ser/Thr protein kinase, since it is known that the isolated cytoplasmic domain can phosphorylate on Ser and Thr residues (Wang et al., 2014; https://doi.org/10.1016/j.jprot.2014.06.009).

Strengths:The work uses logical and well-controlled approaches throughout, and is clear and convincing in most areas, linking data from IPs, kinase assays (including clear 32P-based biochemistry), HD-MX data (from non-phosphorylated EFR) structural biology, oxidative burst data and infectivity assays. Repetitions and statistical analysis all appear appropriate.Overall, the work builds a convincing story and the discussion does a clear job of explaining the potential impact of these findings (and perhaps an explanation of why so many Arabidopsis kinases are 'pseudokinases', including XPS1 and XIIa6, where this is shown explicitly).Weaknesses:No major weaknesses are noted from reviewing the data and the paper follows a logical course built on solid foundations; the use of Tables to explain various experimental data pertinent to the reported studies is appreciated.(1) The use of a, b,c, d in Figures 2C and 3C etc is confusing to this referee, and is now addressed in the latest version(2) The debate about kinase v pseudokinases is well over a decade old. For non-experts, the kinase alignments/issues raised are in PMID: 23863165 and might prove useful if cited.

We have cited the suggested reference in the second paragraph of the discussion.

(3) Early on in the paper, the concept of kinases and pseudokinases related to R-spine (and extended R-spine) stability and regulation really needs to be more adequately introduced to explain what comes next; e.g. some of the key work in this area for RAF and Tyr kinases where mutual F-helix Phe amino acid changes are evaluated (conceptually similar to this study of the E-helix Tyr to Phe changes in EFR) should be cited (PMID: 17095602, 24567368 and 26925779).

As an alternative, we have amended the text in several places to focus on conformational toggling between active/inactive states rather than R-spine stability. We think that this keeps the message of our manuscript focused. We hope that the reviewer finds this acceptable.

(4) In my version, some of the experimental text is also currently in the wrong order (and no page numbers, so hard for me to state exactly where in the manuscript); However, I am certain that Figure 2C is mentioned in the text when the data are actually shown in Figure 3C for the EFR-SSAA protein.

Indeed, some references to Figure 2 in the text were incorrect. We have corrected these. References in the text to Figure 3 and the data reported therein are correct.

(5) Tyr 156 in PKA is not shown in Supplement 1, 2A as suggested in the text; for readers, it will be important to show the alignment of the Tyr residue in other kinases; this has been updated in the second version. Although it is clearly challenging to generate phosphorylated EFR (seemingly through Codon-expansion here?), it appears unlikely that a phosphorylated EFR protein, even semi-pure, couldn't have been assayed to test the idea that the phosphorylation drives/supports downstream signaling. What about a DD or EE mutation, as commonly used (perhaps over-used) in MEK-type studies?

Our aim with codon expansion was to generate recombinant protein carrying high-stoichiometry phosphorylation at sites which we have previously documented to be required for downstream signaling (Macho *et al*., 2014; Bender *et al*., 2021). We additionally demonstrated previously that a DD mutant of the activation loop sites in EFR does not fully complement the efr-1 mutant (Bender et al., 2021), suggesting that the Asp mutations are not good phospho-mimics in this context. We therefore did not generate DD or EE mutations for in vitro studies.

Impact:The work is an important new step in the huge amount of follow-up work needed to examine how kinases and pseudokinases 'talk' to each other in (especially) the plant kingdom, where significant genetic expansions have occurred. The broader impact is that we might understand better how to manipulate signaling for the benefit of plants and mankind; as the authors suggest, their study is a natural progression both of their own work, and the kingdom-wide study of the Kannan group.
**Reviewer #3 (Public Review):**
The study presents strong evidence for allosteric activation of plant receptor kinases, which enhances our understanding of the non-catalytic mechanisms employed by this large family of receptors.Plant receptor kinases (RKs) play a critical role in transducing extracellular signals. The activation of RKs involves homo- or heterodimerization of the RKs, and it is believed that mutual phosphorylation of their intracellular kinase domains initiates downstream signaling. However, this model faces a challenge in cases where the kinase domain exhibits pseudokinase characteristics. In their recent study, Mühlenbeck et al. reveal the non-catalytic activation mechanisms of the EFR-BAK1 complex in plant receptor kinase signaling. Specifically, they aimed to determine that the EFR kinase domain activates BAK1 not through its kinase activity, but rather by utilizing a "conformational toggle" mechanism to enter an active-like state, enabling allosteric trans-activation of BAK1. The study sought to elucidate the structural elements and mutations of EFR that affect this conformational switch, as well as explore the implications for immune signaling in plants. To investigate the activation mechanisms of the EFR-BAK1 complex, the research team employed a combination of mutational analysis, structural studies, and hydrogen-deuterium exchange mass spectrometry (HDX-MS) analysis. For instance, through HDX-MS analysis, Mühlenbeck et al. discovered that the EFR (Y836F) mutation impairs the accessibility of the active-like conformation. On the other hand, they identified the EFR (F761H) mutation as a potent intragenic suppressor capable of stabilizing the active-like conformation, highlighting the pivotal role of allosteric regulation in BAK1 kinase activation. The data obtained from this methodology strengthens their major conclusion. Moreover, the researchers propose that the allosteric activation mechanism may extend beyond the EFR-BAK1 complex, as it may also be partially conserved in the Arabidopsis LRR-RK XIIa kinases. This suggests a broader role for non-catalytic mechanisms in plant RK signaling.The allosteric activation mechanism was demonstrated for receptor tyrosine kinases (RTKs) many years ago. A similar mechanism has been suggested for the activation of plant RKs, but experimental evidence for this conclusion is lacking. Data in this study represent a significant advancement in our understanding of non-catalytic mechanisms in plant RK signaling. By shedding light on the allosteric regulation of BAK1, the study provides a new paradigm for future research in this area.
**Recommendations for the authors:**

**Reviewer #1 (Recommendations For The Authors):**
The authors have considered points 1-5 raised in my initial review and the revised manuscript contains a more balanced discussion and limitation section. No additional experiments have been performed to substantiate the envisioned allosteric activation mechanism of the co-receptor kinase BAK1 by the receptor EFR. I rewrote the public statement accordingly.
**Reviewer #2 (Recommendations For The Authors):**
Thanks for responding to my comments.
**Reviewer #3 (Recommendations For The Authors):**
The revised manuscript has fully addressed my previous concerns and is now suitable for publication in eLife.